# Endogenous retroviruses are a source of enhancers with oncogenic potential in acute myeloid leukaemia

Özgen Deniz 1,2✉, Mamataz Ahmed1,2, Christopher D. Todd 1,2,5, Ana Rio-Machin 2,3, Mark A. Dawson 4 & Miguel R. Branco 1,2✉

Acute myeloid leukemia (AML) is characterised by a series of genetic and epigenetic alterations that result in deregulation of transcriptional networks. One understudied source of transcriptional regulators are transposable elements (TEs), whose aberrant usage could contribute to oncogenic transcriptional circuits. However, the regulatory influence of TEs and their links to AML pathogenesis remain unexplored. Here we identify six endogenous retrovirus (ERV) families with AML-associated enhancer chromatin signatures that are enriched in binding of key regulators of hematopoiesis and AML pathogenesis. Using both locus-specific genetic editing and simultaneous epigenetic silencing of multiple ERVs, we demonstrate that ERV deregulation directly alters the expression of adjacent genes in AML. Strikingly, deletion or epigenetic silencing of an ERV-derived enhancer suppresses cell growth by inducing apoptosis in leukemia cell lines. This work reveals that ERVs are a previously unappreciated source of AML enhancers that may be exploited by cancer cells to help drive tumour heterogeneity and evolution.

---

[1] Blizard Institute, Barts and The London School of Medicine and Dentistry, QMUL, London E1 2AT, UK. [2] Centre for Genomic Health, Life Sciences Institute, QMUL, London E1 2AB, UK. [3] Centre for Haemato-Oncology, Barts Cancer Institute, QMUL, London EC1M 6BQ, UK. [4] Cancer Research Division, Peter MacCallum Cancer Center, Melbourne, Victoria 3002, Australia. [5] Present address: Epigenetics Programme, Babraham Institute, Cambridge CB22 3AT, UK.
✉email: o.deniz@qmul.ac.uk; m.branco@qmul.ac.uk

Acute myeloid leukaemia (AML) is characterised by clonal proliferation of immature myeloid cells. AML is highly heterogeneous at both the genetic and biological level, and individuals with AML accumulate a wide variety of genetic alterations that affect signalling pathways, transcription factors (TFs) and epigenetic modifiers[1]. In addition to genetic alterations, epigenetic processes have been shown to play key, and sometimes independent, dynamic roles in the molecular pathogenesis of AML[2,3]. For instance, altered chromatin landscapes, including DNA methylation[4], histone modifications and chromatin accessibility[5,6], are characteristics of AML subtypes. Genetic and epigenetic perturbations often target transcriptional regulatory networks, leading to dysregulation of transcriptional programmes in AML and conferring a selective advantage[5,7]. During malignant transformation, leukaemia cells undergo continuous genetic and epigenetic diversification, thereby increasing inter- and intra-patient tumour heterogeneity[3,8], which directly reflects the complexity of leukaemic transcriptional programmes. One key component of transcriptional networks are transposable elements (TEs), which provide a rich source of tissue-specific cis-regulatory DNA sequences[9]. Despite extensive functional genomic analyses of AML, crucially the contribution of TEs to this disease is currently unknown.

TEs have integrated into the human genome at different times throughout evolution and currently comprise around half of our genome. Based on their evolutionary origins, TEs vary with regard to their DNA structure. For instance, long terminal repeat (LTR) retrotransposons, which include endogenous retroviruses (ERVs), are composed of two LTRs that flank an internal retrovirus-derived coding region[10]. However, LTRs frequently recombine, leaving the majority of ERV elements as intact solitary LTRs, which contain functional cis-regulatory DNA sequences[11,12]. Therefore, ERVs are fixed in our genome, but still maintain intrinsic regulatory capacity. Consistent with this, genome-wide assays have demonstrated that numerous LTR sequences carry hallmarks of active regulatory elements[13–20]. In a few instances, loss-of-function experiments have provided compelling evidence of LTR contribution to host gene regulation and cellular function in erythropoiesis[21], innate immunity[18], pregnancy[22] and fertility[23].

Various studies have documented widespread epigenetic and transcriptional deregulation of TEs in several cancer types, raising the possibility that TE-derived regulatory elements may be exploited to promote tumorigenesis[24,25]. Indeed, activation of LTR-based promoters initiates cancer-specific chimeric transcripts in Hodgkin lymphoma, melanoma and diffuse large B-cell lymphoma, amongst others[24,26,27]. However, studies to date have been centred on LTR promoter activity and its potential function as enhancer remains unexplored in human malignancies. Through the direct physical interactions with promoters, enhancers are especially important to regulate gene expression in a cell type-, temporal- and differentiation-stage-specific manner, all of which are essential for maintaining normal haematopoiesis. Indeed, dysregulation of specific enhancers, as well as global epigenetic disruption of the enhancer landscape have been shown to play critical roles in AML pathogenesis[28–30]. In this context, TEs are an ideal source of novel regulatory regions that could be co-opted in order to promote expression of genes essential for leukaemic transformation and evolution in AML.

Here we use epigenomic and transcriptomic data from primary AML samples and leukaemia cell lines to explore the potential regulatory roles of TEs in AML. We identify six ERV/LTR families with regulatory potential that harbour enhancer-specific epigenetic signatures and bind TFs that play key roles in haematopoiesis and in the pathogenesis of AML. Moreover, deletion of individual ERVs and epigenetic inactivation of an entire ERV family demonstrate their direct roles in gene regulation. Strikingly, we find that either genetic or epigenetic perturbation of a single ERV-derived enhancer element leads to impaired cell growth by modulating expression of the APOC1 gene, suggesting that the activation of this particular ERV has a driving role in leukaemia cell phenotype.

## Results

**Identification of putative AML-specific regulatory TEs.** To identify putative regulatory TEs, we generated DNase-seq data from three commonly used AML cell lines with different genetic and cytogenetic backgrounds: HL-60, MOLM-13 and OCI-AML3. In addition, we analysed DNase-seq data from 32 AML samples generated by the Blueprint epigenome project[6], and compared them with data from differentiated myeloid cells (macrophages and monocytes) from the same consortium (Fig. 1a). We overlapped DNase-hypersensitive sites (DHSs) with the complete Repeatmasker annotation and compared the DHS frequency at each repeat family with random controls (Supplementary Data 1). We identified twelve repeat families that were enriched for DHS-associated copies in at least one of the AML cell lines and in 10% or more of the AML samples (Fig. 1b). Five of these repeat families (three of which are not TEs) were highly enriched across all samples, including macrophages and monocytes, as well as mobilised CD34+ cells (data from the Roadmap epigenomics project), suggesting little cell specificity. The remaining seven families displayed more variability between AML samples and, notably, tended to display little or no enrichment in differentiated myeloid cells (Fig. 1b). Nearly all families were also DHS-enriched in CD34+ cells, suggesting an association with a stem-cell state, which may be exploited by cancer cells to promote cell proliferation and survival. In contrast, the DHS enrichment of LTR2B elements appeared to be AML-specific and therefore associated only with the disease state. Analysis of an independent dataset of 32 AML samples from the Bonifer lab[5] confirmed the DHS enrichment at all of the above families, and identified additional weaker associations, including with several Alu subfamilies (Supplementary Fig. 1A). For stringency, we focused on families that were DHS-enriched in both datasets, all of which are LTRs from ERVs: LTR2B, LTR2C, LTR5B, LTR5_Hs, LTR12C and LTR13A. We excluded the internal portion of HERVK (HERVK-int) because its enrichment was largely due to its LTRs (LTR5B, LTR5_Hs; Supplementary Fig. 1B). We will collectively refer to the six selected ERV families as 'AML DHS-associated repeats' (A-DARs). The oldest A-DARs (LTR5B and LTR13A) date back to the common ancestor between hominoids and old-world monkeys, whereas the youngest (LTR5_Hs) are human-specific[31].

The DNase-seq profiles across each ERV displayed a consistent pattern for elements of the same family in AML cell lines (less evident for LTR2C), suggestive of TF-binding events within these ERVs (Fig. 1c displays OCI-AML3 profiles). This pattern was also notable in primary AML cells, albeit variable between samples (Supplementary Fig. 1C), reflecting the heterogeneity of this disease. Out of a total of 4811 A-DAR elements, 80–661 (median 263) overlapped a DHS in AML samples from the Blueprint dataset and 223–1349 (median 508) in the Assi et al. dataset. As heterogeneity in AML is partly driven by genetics, we hypothesised that variation in DHS frequency at A-DARs could reflect distinct mutational profiles. To test this, we measured inter-sample correlations in the DHS patterns of A-DARs, which revealed distinct clusters associated with the mutational profile in AML patient samples (Supplementary Fig. 2A). Although there was no strict association with particular AML subtypes, we found that samples with NPM1 mutations were better inter-correlated

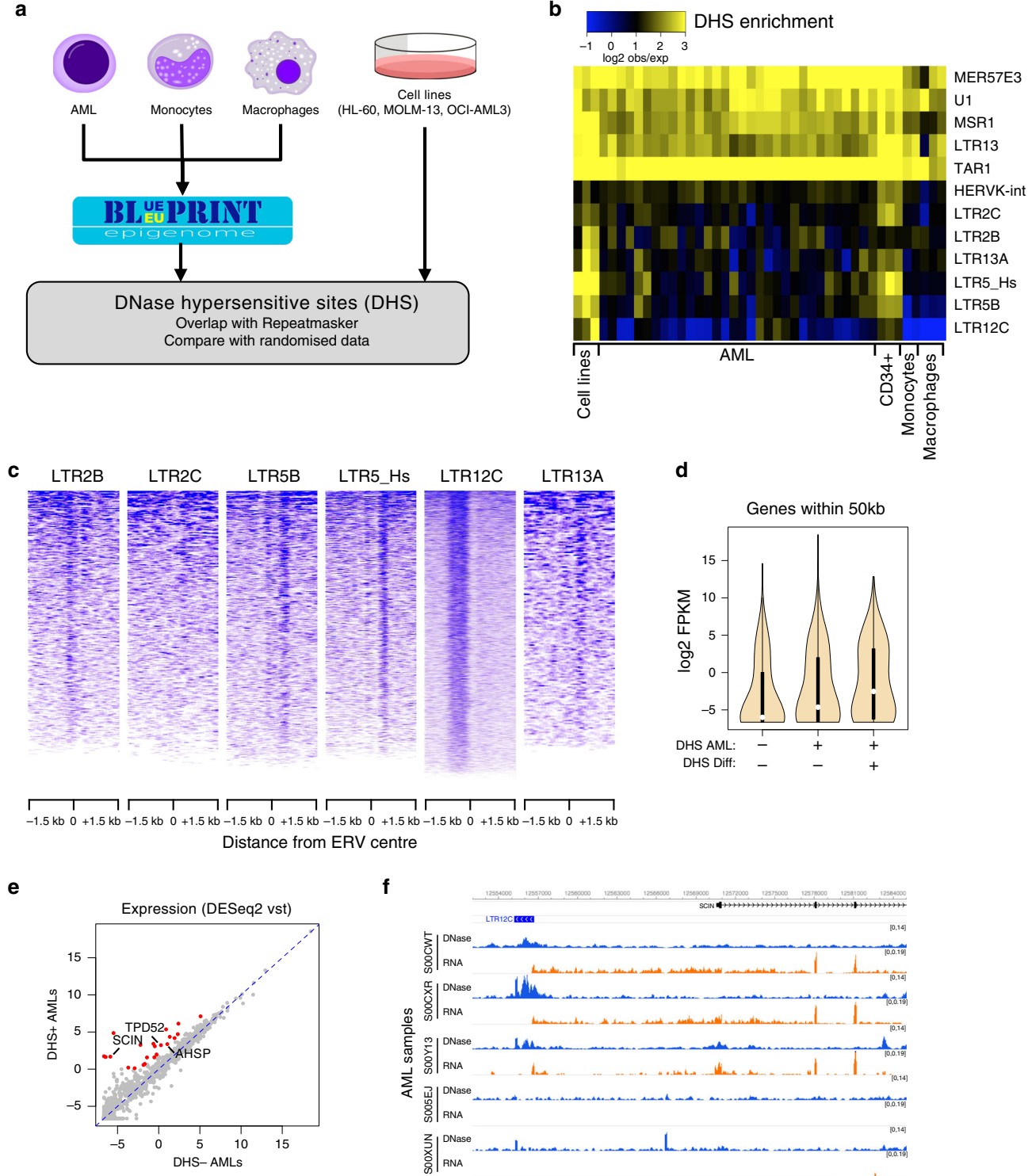

**Fig. 1 ERVs with regulatory potential are activated in AML. a** Schematic of the strategy to detect repeat families associated with open chromatin in AML (hematopoietic cells' credit: A. Rad and M. Häggström; CC-BY-SA-3.0 licence). **b** Heatmap of the observed/expected enrichment for DHSs in selected repeat families. Cell lines are presented in the following order: HL-60, MOLM-13 and OCI-AML3. **c** DNase-seq profile across all elements of each AML DHS-associated repeat (A-DAR) families in OCI-AML3. **d** Gene expression average across all Blueprint AML samples for genes within 50 kb of A-DARs with or without a DHS in AML and/or in differentiated cells (boxes indicate first, second (median) and third quartiles; whiskers indicate data within 1.5× of the interquartile range). **e** For each gene lying near an A-DAR element, we compared its expression in AML samples ($n = 26$) where the respective ERV has a DHS, versus AML samples where the DHS is absent. Expression values were normalised using the variance-stabilising transformation (vst, log2 scale) in DESeq2. Highlighted are genes with >4-fold difference and vst > 0. **f** Example of a gene (*SCIN*) that displays a strict correlation between its expression (orange) and the presence of a DHS peak (blue) at a nearby LTR12C element in different AML samples.

than those without (Supplementary Fig. 2B). The same was true for samples with *FLT3*-ITD and *DNMT3A* mutations, which frequently co-occur with *NPM1* mutations, as well as those with *CEBPA* mutations (Supplementary Fig. 2B). Specific mutations may therefore contribute to ERV activation in AML, although other characteristics of the malignancy are also likely to affect them.

**A-DAR chromatin status correlates with nearby gene expression**. To test whether A-DARs were associated with gene activation, we analysed matching DNase-seq and RNA-seq data from the Blueprint consortium (n samples: 27 AML, 6 macrophages and 8 monocytes). ERVs can not only affect the expression of proximal genes, but also act at a distance via long-range interactions in 3D space[19,32]. However, long-range interactions display substantial cell specificity, namely within the haematopoietic system[33]. Given the heterogeneity between AML samples and the lack of matching Hi–C data, we stringently focused our analysis on genes within 50 kb of an ERV from the selected families. Genes close to A-DAR elements with DHS in two or more AML samples displayed higher expression levels than those close to A-DAR elements without DHS (Fig. 1d). This was more pronounced for ERVs with DHS also present in differentiated cells. Even though such bulk correlations are only suggestive of a regulatory role of ERVs, we found individual elements with strong supporting evidence for their regulatory activity, as the expression levels of their adjacent genes were greater than four-fold higher in AML samples with DHS at a given ERV, versus those without (Fig. 1e; see also Supplementary Data 7). This included a strict correlation between chromatin accessibility at a LTR12C element and the expression of the *SCIN* gene (Fig. 1F). Notably, low *SCIN* expression is associated with an adverse AML prognosis[34]. Two other genes of interest for which expression also correlates with a DHS at nearby ERVs are *TPD52* and *AHSP*, whose overexpression in AML is predictive of poor and favourable outcomes, respectively[35,36]. These data suggest that at least some A-DAR elements gain gene-regulatory activity in AML, which correlates with disease outcomes.

**A-DARs bear the chromatin signatures of enhancer elements**. DNase hypersensitivity is associated with both active gene promoters and distal enhancers. LTR12C elements, for example, were previously shown to frequently act as alternative gene promoters in different cell types, including hepatocellular carcinoma[37] and cell lines treated with DNMT and HDAC inhibitors[38]. In contrast, LTR5_Hs (HERVK) elements appear to mainly act as distal enhancer elements in embryonic carcinoma cells and stem cells[19,20]. We therefore aimed to establish whether A-DARs could act as promoters and/or enhancers in AML.

To test for gene promoter activity, we performed de novo transcriptome assembly in AML samples and differentiated myeloid cells, and calculated the number of spliced transcripts for which the transcriptional start site (TSS) overlapped an A-DAR element. AML samples displayed 31–53 such transcripts, whereas differentiated cells had 20–28, most of which emanated from LTR12C elements (Fig. 2a). We identified 82 spliced transcripts that were present in two or more AML samples, but were absent in differentiated cells (Supplementary Data 2). Most of these were short transcripts and only 28 had evidence of splicing into exons of annotated genes. RT-qPCR and/or CAGE analyses on primary samples would be required to validate such alternative TSSs emerging from A-DAR elements, especially given that only a subset is supported by GENCODE or FANTOM5 annotations (Supplementary Data 2). Nevertheless, one notable example involved a LTR2C element active in a subset of AMLs,

which acted as a non-reference promoter for *SAGE1* (Fig. 2b), a known cancer/testis antigen[39,40]. Another example is an LTR2B element that is active in the majority of AML samples, and is an annotated promoter of the *RHEX* gene. *RHEX* regulates erythroid cell expansion[41], and is highly expressed in AML (Blueprint Data Analysis Portal, http://blueprint-data.bsc.es).

We then asked whether A-DARs are marked by promoter- or enhancer-associated histone modifications. Using ChIP-seq data from the Blueprint consortium (*n* samples: 29 AML, 7 macrophages and 8 monocytes), we first plotted the percentage of elements from each ERV family that were marked by H3K27ac, H3K4me1, H3K4me3 or H3K9me3 in AML and differentiated myeloid cells (Supplementary Fig. 3A). Notably, in AML samples, an average 5.7–15.2% of elements from each family overlapped H3K4me1 peaks, a mark predominantly associated with poised and active enhancers. This was substantially higher than the fraction overlapping with the active promoter mark H3K4me3 (1.3–3.4%). Indeed, a more detailed analysis of histone modification patterns at A-DAR elements showed that H3K4me1 is either found in conjunction with H3K27ac (active enhancers), or on its own (primed enhancers), but is rarely found together with H3K4me3 (Fig. 2c, Supplementary Fig. 3B). Clustering analysis of these patterns demonstrated that while some elements within a family bear active marks in both AML and differentiated cells, a substantial portion (10–37%, depending on the family, median 20%) displays enhancer-like profiles only in AML samples (Fig. 2c, Supplementary Fig. 3B). ChIP-seq profiles confirmed that these AML-specific elements had elevated H3K4me1 and H3K27ac in AML when compared with differentiated cells (Fig. 2d). A total of 1122 and 411 A-DAR elements were marked by H3K4me1 and H3K27ac, respectively (333 had both marks), in at least two AML samples. A-DARs are therefore frequently associated with enhancer-like profiles in AML.

To test whether myeloid leukaemia cell lines could be used to dissect the putative enhancer roles of A-DARs, we performed H3K27ac ChIP-seq on HL-60, MOLM-13, OCI-AML3 and K562 cells, and compared patterns with those seen in AML samples. A-DAR elements that overlap H3K27ac peaks in AML samples were also frequently associated with this mark in cell lines (Fig. 2E). A ChromHMM annotation for K562 cells from ENCODE further supported that these elements often bear enhancer signatures (Fig. 2e). It is worth noting that there is substantial variation in H3K27ac enrichment of A-DARs among cell lines, much like in primary AML samples. Nonetheless, example loci show that H3K27ac deposition at A-DAR elements in cell lines can recapitulate primary AML data (Fig. 2f), opening up the opportunity to functionally test for enhancer activity of these loci in cell lines.

**A-DARs bind AML-related TFs**. Previous ChIP-seq or motif analyses had identified several TFs associated with the ERV families identified here[15,17]. These included haematopoiesis- and AML-related TFs such as TAL1, SPI1, GATA2 and ARNT, amongst others. To confirm and extend these observations, we first performed our own analysis of TF ChIP-seq data from K562 cells (ENCODE consortium). Our comparison with AML data above gave us confidence that K562 cells were an adequate model to study TF-binding patterns at A-DARs. We analysed all TF ChIP-seq peak data available from ENCODE and selected TFs that are bound to at least 5% of the elements in a given ERV family, in a statistically significant manner, yielding a list of 217 TFs (Fig. 3a; Supplementary Data 3). The vast majority of these TFs were found to be expressed in AML samples (198 had higher expression than TBP), and many of them are involved in hae-matopoietic gene regulation and/or in the aetiology of AML,

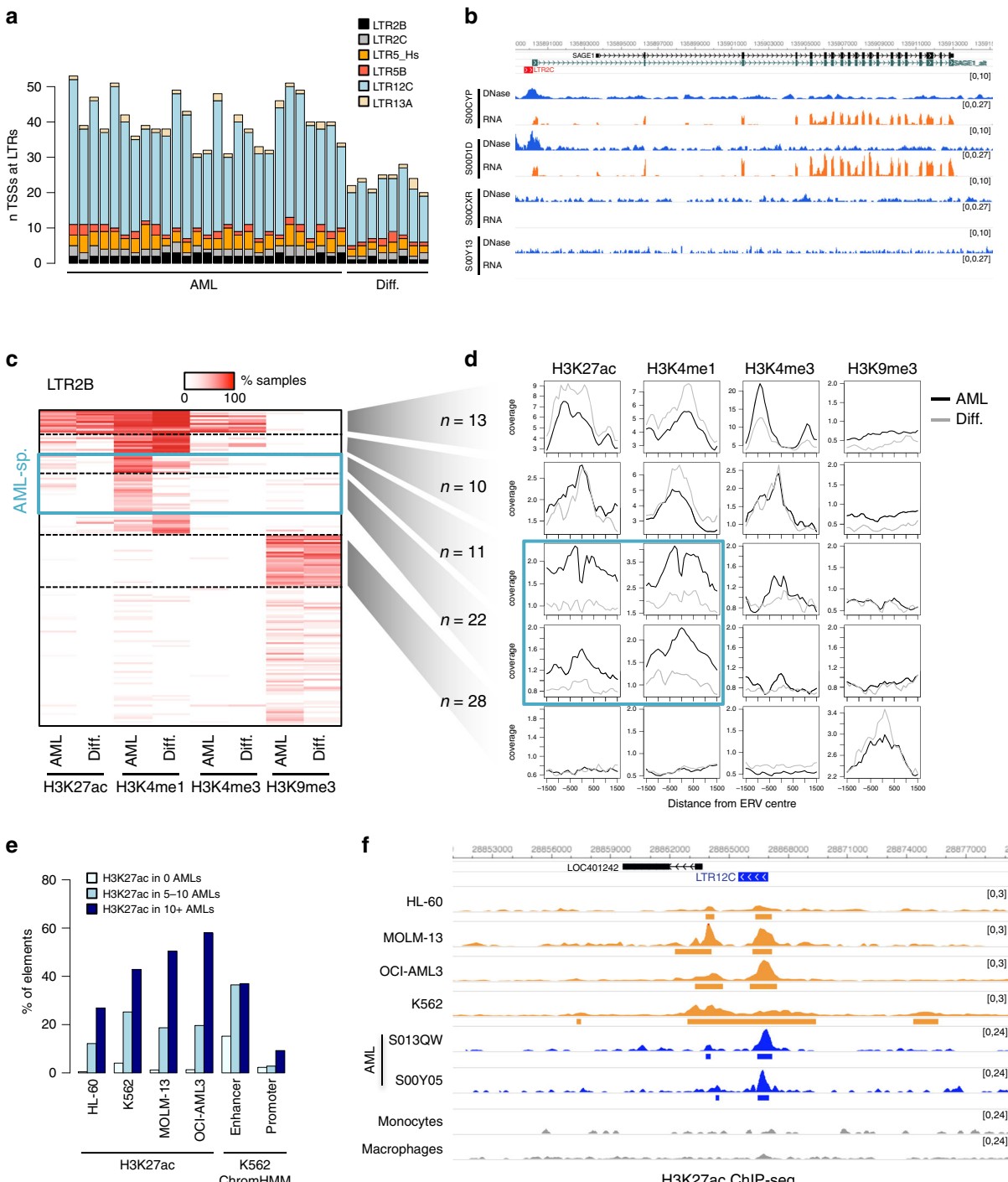

**Fig. 2 A-DARs bear signatures of enhancer elements. a** Number of transcriptional start sites of spliced transcripts that overlap with A-DAR elements in AML or differentiated myeloid cells. **b** Example of a LTR12C element that generates an alternative promoter that drives the expression of *SAGE1* in AML samples where this element is active. **c** Heatmap of overlap between LTR2B elements and histone modification peaks. Colour intensity represents the percentage of AML or differentiated cell samples where overlap is observed. Dashed lines segregate clusters identified by k-means clustering. **d** Average ChIP-seq profiles for LTR2B elements within specific clusters defined in (**c**). Blue boxes highlight two clusters where H3K4me1 and H3K27ac levels are higher in AML compared with differentiated cells. **e** Percentage of A-DAR elements that overlap H3K27ac peaks in different cell lines, or that are classified as enhancers or promoters in ChromHMM data from K562 cells. A-DAR elements were subdivided according to the number of AML samples displaying overlap with H3K27ac. **f** Example of a LTR13A element where cell lines reproduce the AML-specific H3K27ac marking observed in AML samples. Peaks called by MACS2 are depicted underneath each track.

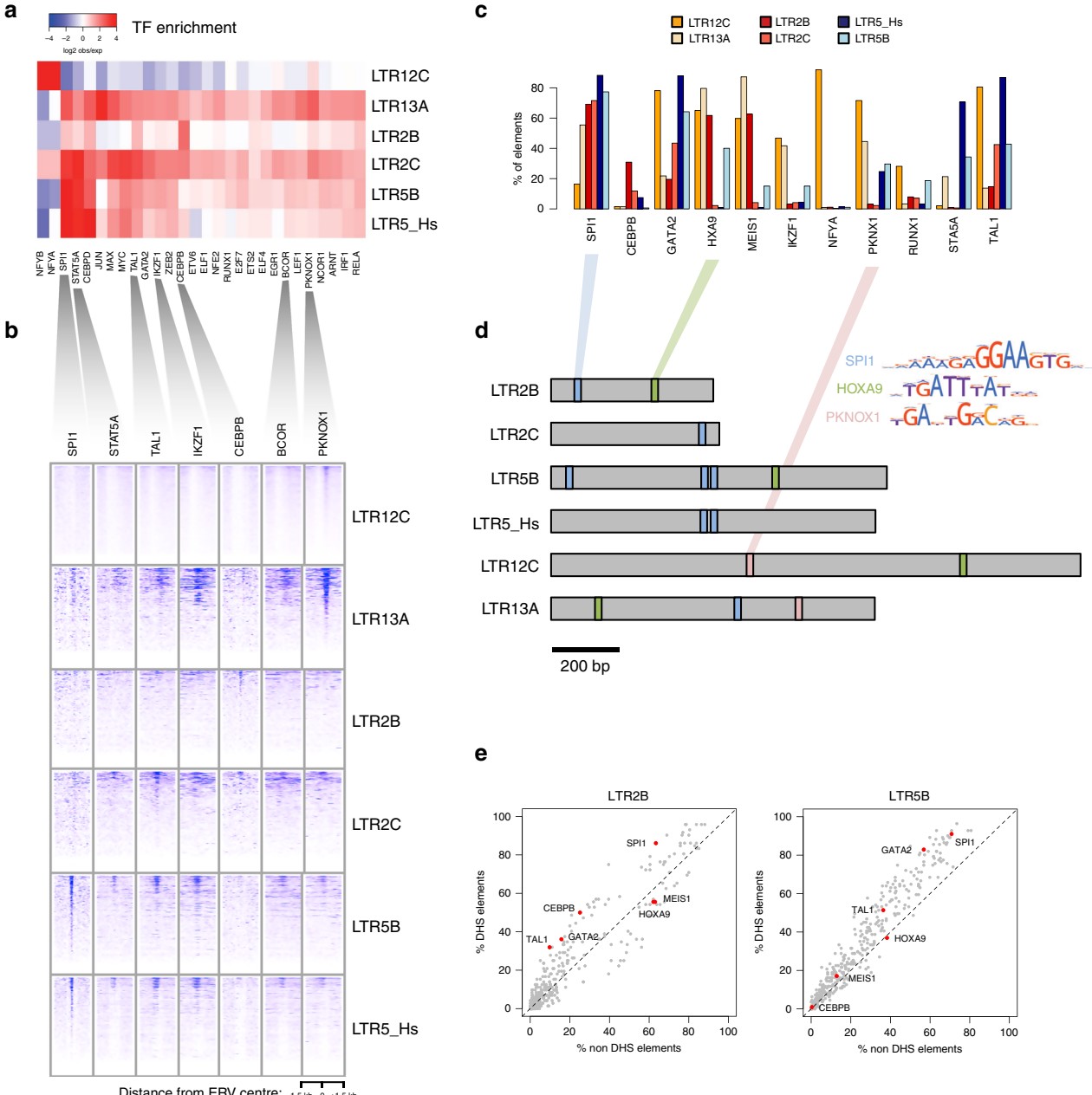

**Fig. 3 A-DARs bind AML-related transcription factors (TFs). a** Heatmap of the observed/expected enrichment for TF-binding sites in K562 cells. **b** ChIP-seq profiles of selected TFs across all elements of each A-DAR family. For each family, elements are displayed in the same order across all TF profiles. **c** Percentage of ERVs from each family bearing a binding motif for the indicated TFs. **d** Location of selected TF motifs at the consensus sequences of each A-DAR family. **e** TF motif frequency at LTR2B and LTR5B elements, comparing those that overlap DHSs with those that do not.

including SPI1, TAL1, IKZF1 and PKNOX1 (Fig. 3a). ChIP-seq profiles of individual elements revealed a localised pattern of TF binding at a subset of elements (Fig. 3B), with different ERV families binding different combinations of TFs. To evaluate TF binding in a primary cell type, we analysed data from CD34+ haematopoietic progenitors, from the BloodChIP database[42]. This revealed clear binding enrichment for FLI1, GATA2, LYL1, RUNX1 and TAL1 in at least one of the ERV families (Supplementary Fig. 4).

We also performed TF motif analysis (Fig. 3c, Supplementary Data 4), which was largely congruent with the ChIP-seq data. Apart from confirming the presence of motifs for SPI1, PKNOX1 and other TFs, in four different ERV families we found enrichment for motifs for HOXA9/MEIS1, co-expression of

which is sufficient to drive leukaemogenesis in mouse models[43]. In line with the high frequency of many of the identified TF motifs, we found that they were present in the consensus sequences of each ERV family (Fig. 3d), suggesting that the respective retroviruses were brought in these motifs within their LTRs upon invasion of the human genome. Finally, we asked whether some TF motifs were responsible for chromatin opening at individual elements. We tested for motif enrichment in elements with DHSs (DHS+) in at least five of the analysed AML samples, when compared with DHS-negative elements (Supplementary Data 5). In four of the ERV families, we identified several enriched motifs (none in LTR2C or LTR13A), such as TAL1 in LTR2B, LTR5_Hs and LTR12C), CEBPB (in LTR2B) and GATA2 (in LTR5B and LTR12C). However, the differences in motif

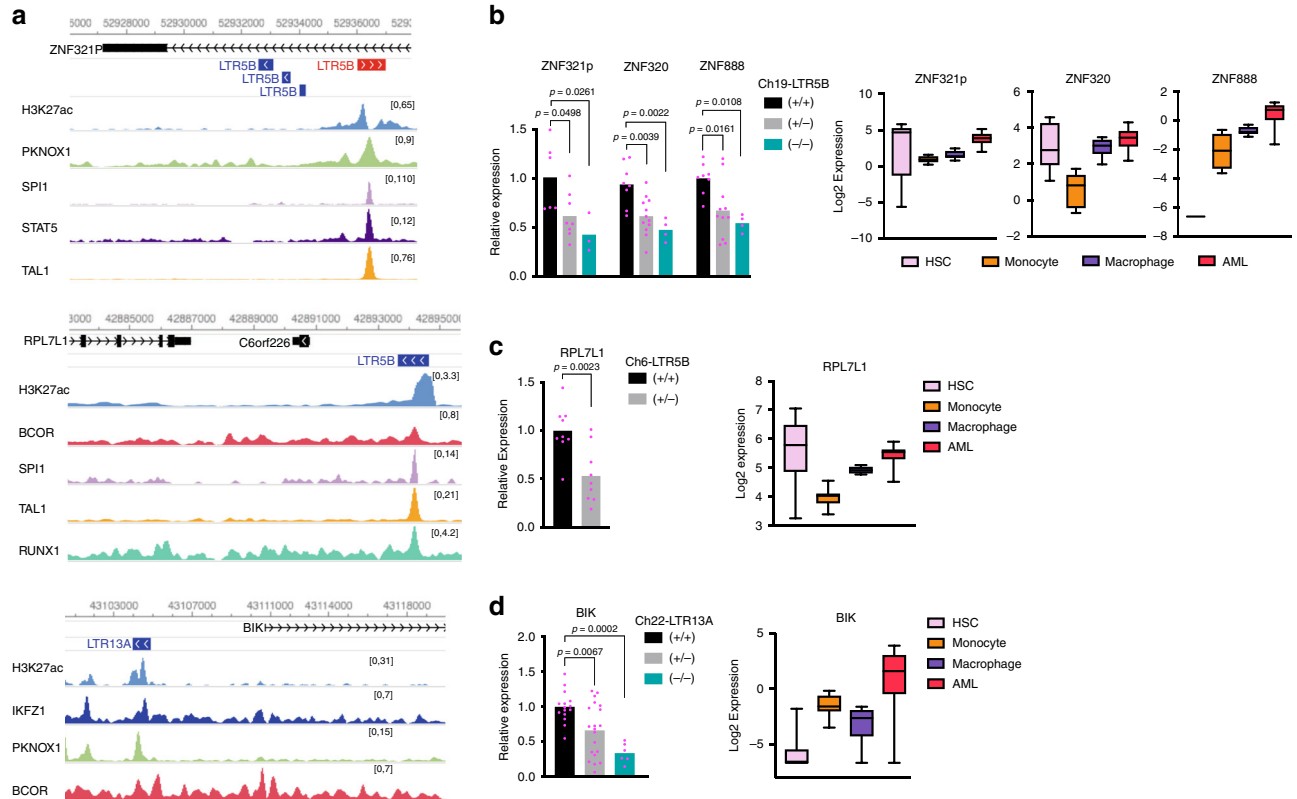

**Fig. 4 Regulatory ERVs modulate host gene expression. a** Genome browser view of three candidate ERVs, showing H3K27ac and TF ChIP-seq tracks in K562 cells. **b–d** Expression of nearby genes (left) in the excision clones of the indicated ERVs. Bars represent mean values. Data points are from multiple samples (collected every 2–4 days) from each independently derived clone, as follows: four samples from 2+/+, 3+/− and 1−/− clones (**b**), 3 samples from 3+/+ and 3+/− clones (c), and 8 samples from 2+/+, 3+/− and 1−/− clones (**d**). P values are from one-way ANOVA with Tukey's multiple-comparison test (**b**, **d**) or two-tailed t test (**c**). Expression of the indicated ERV vicinity genes (right) in HSC (n = 6), monocyte (n = 8), macrophage (n = 6), and AML (n = 27) samples (boxes indicate first, second (median) and third quartiles; whiskers indicate data within 1.5× of the interquartile range). Source data are provided as a Source Data file for (**b**–**d**).

frequency between DHS+ and DHS− elements were modest, making TF motifs poor discriminators of these two groups (Fig. 3e, Supplementary Fig. 5). For example, even though SPI1 binding motif is present in the majority of DHS+ elements, a large portion of non-DHS elements also harbour this motif (Fig. 3e). This suggests that other factors play a role in determining LTR regulatory potential, in line with our previous observations in mouse stem cells[44].

These analyses suggest that the potential regulatory activity at particular ERV families in AML is likely driven by the binding of haematopoiesis-associated TFs, which are either upregulated in AML or whose binding sites become accessible in AML through epigenetic alterations.

**Genetic excision of A-DAR elements interferes with host gene expression.** To test for causal roles of enhancer-like A-DAR elements in gene regulation, we used CRISPR–Cas9 to delete three candidate ERVs (Supplementary Fig. 6). The selected ERVs are enriched in H3K27ac, bound by multiple haematopoiesis-associated TFs in K562 cells (Fig. 4a), and overlap DHSs in multiple AML samples, but not in monocytes or macrophages (Supplementary Fig. 7). We generated clones with heterozygous or homozygous deletions of these ERVs in K562 cells, and measured the expression of associated genes in multiple clones. Other leukaemia cell lines (HL-60, OCI-AML3 and MOLM-13) proved more refractory to genetic deletion, due to the low efficiency of Cas9 delivery and single-cell expansion.

One of the deleted loci was a LTR5B element located in the first intron of *ZNF321P*, which is bound by PKNOX1, SPI1, STAT5 and TAL1 (Fig. 4a, top). Deletion of this element led to a significant decrease in *ZNF321P* expression and also affected the expression of two other nearby genes, *ZNF320* and *ZNF888* (Fig. 4b, left). Notably, all three genes display higher expression in AML samples when compared with monocytes and macrophages (Fig. 4b, right). Interestingly, *ZNF320* is also upregulated in multiple cancer types[45]. ZNF320 is a member of the Krüppel-associated box (KRAB) domain zinc finger family and predominantly binds LTR14A and LTR14B elements[46], suggesting a potential role in ERV silencing. Heterozygous deletion of another LTR5B element, bound by BCOR, SPI1, TAL1 and RUNX1 (Fig. 4a, middle), reduced the expression of Ribosomal Protein L7 Like 1 (*RPL7L1*) (Fig. 4c, left), which is upregulated in AML when compared with differentiated myeloid cells (Fig. 4c, right). Notably, this specific LTR5B contains a single-nucleotide polymorphism (SNP) for which the minor allele (highest population frequency of 0.41) disrupts a MAFK-binding motif (Supplementary Fig. 7C). Using data from the GTEx project, we found that the minor allele was associated with lower *RPL7L1* expression in whole blood (Supplementary Fig. 7C), suggesting that the MAFK motif is important for *RPL7L1* expression. The third deleted locus was an LTR13A element located in the vicinity of *BCL2*-interacting killer (*BIK*), and is enriched for IKFZ1, PKNOX1 and BCOR binding (Fig. 4a, bottom). Excision of this particular element led to around threefold reduction in *BIK* expression (Fig. 4d, left), which is higher in AML samples when

compared with other haematopoietic cell types (Fig. 4d, right). This LTR13A also contains a SNP, where the minor allele (highest population frequency of 0.5) is a critical residue in a RUNX1-binding site, but that was not associated with any significant differences in *BIK* expression in whole blood (Supplementary Fig. 7E).

Overall, CRISPR-mediated genetic deletion assays demonstrate a direct role of individual A-DAR elements in gene regulation in K562 cells. Moreover, DHSs within the candidate ERVs and high expression of their associated genes in AML patients provide strong evidence for their regulatory activation in vivo.

**Inactivation of LTR2B elements leads to growth suppression.** To test the regulatory function of multiple A-DAR elements simultaneously, we next sought to epigenetically silence one ERV family by CRISPR interference (CRISPRi) using a catalytically dead Cas9 (dCas9) fused to the KRAB transcriptional repressor protein. We targeted the LTR2B family, which was the only one with AML-specific DHS enrichment and no enrichment in CD34 + cells (Fig. 1b), suggesting a more cancer-specific role than other A-DARs. We designed 4 sgRNAs targeting the most conservative regions of the LTR2B family, predicted to recognise around 217 copies (68%). Our LTR2B sgRNAs are also predicted to target copies of highly related LTR2 family (71 copies, 8%). To determine dCas9 specificity on a genome-wide scale, we performed dCas9 ChIP-seq in K562 cell lines expressing LTR2B sgRNAs or empty vector. We detected 395 dCas9 peaks in cells with LTR2B sgRNAs (and none in control cells), 187 of which were associated with LTR2B elements, and 90 with LTR2 elements (Fig. 5a, b). The remaining 118 peaks (Fig. 5b) were included in downstream analyses to evaluate putative off-target effects. We performed H3K27ac and H3K9me3 ChIP-seq in the same cells to assess the epigenetic changes imparted by CRISPRi. We quantified the ratio in histone modification levels at dCas9 peaks between cells expressing LTR2B sgRNAs and those with the empty vector control. As expected, upon CRISPRi in K562 cells, we observed a reduction of H3K27ac signal and/or gain of H3K9me3 signal at most loci bound by dCas9, demonstrating effective epigenetic editing (Fig. 5c, d). Notably, LTR2B/LTR2 target sites generally underwent more pronounced changes in H3K27ac and H3K9me3 levels when compared with off-target sites. Changes in histone modification levels upon CRISPRi were further confirmed by ChIP-qPCR at LTR2B elements (Supplementary Fig. 8A). In OCI-AML3 cells, we observed a similar trend in epigenetic alterations upon CRISPRi, albeit to a lesser extent than in K562 cells (Supplementary Fig. 8B, C).

Intriguingly, proliferation assays showed that epigenetic silencing of LTR2B and LTR2 elements by CRISPRi significantly suppressed cell proliferation in both K562 and OCI-AML3 cell lines (Fig. 5e). To test the impact of LTR2B and LTR2 inactivation on the host transcriptome, and gain insights into the mechanism underlying impaired cell growth, we performed RNA-seq in both cell lines (Fig. 5f; Supplementary Fig. 8D). We identified a total of 58 and 99 differentially expressed genes in K562 and OCI-AML3 cells, respectively (Supplementary Data 6). To elucidate the direct effects of CRISPRi, we focused on genes that are within 50 kb of a dCas9 peak and found 15 and 6 differentially expressed genes (in K562 and OCI-AML3 cells, respectively), all but one of which were downregulated. Only one of these genes (*BIK*), which was downregulated in OCI-AML3, was associated with an off-target dCas9 peak. The remaining genes were associated with 15 different LTR2B/LTR2 elements. Four of these elements were intronic, and thus we cannot exclude the possibility that dCas9 binding interfered with transcriptional elongation[47]. In some instances, the LTR2B/LTR2 element was

very close to the promoter of the affected gene, such that silencing could have resulted from H3K9me3 spreading. We therefore performed genetic deletion of one of these elements, which also led to a decrease in expression of the adjacent *ZNF611* gene, albeit to a lesser extent than by CRISPRi (Supplementary Fig. 8E). Several genes displayed decreased expression in both cell lines (Fig. 5g), although only apolipoprotein C1 (*APOC1*) reached statistical significance in both contexts. Notably, five apolipoprotein genes were downregulated in at least one of the cell lines. *APOC1*, *APOC2*, *APOC4–APOC2* and *APOE* lie within a cluster on chromosome 19, and may all be controlled by the same LTR2 element, located upstream of *APOC1*. On the other hand, *APOL1* is on chromosome 22 and close to an LTR2B insertion. Given the key roles that lipid metabolism plays in supporting cancer cell survival[48], the coordinated downregulation of apolipoprotein genes could underpin the reduced cell growth observed upon silencing of LTR2B/LTR2 elements in leukaemia cell lines.

Overall, these data show that a subset of LTR2B and LTR2 elements act as key gene regulators in leukaemia cell lines, and that their epigenetic silencing impairs cell growth, providing evidence for a functional role in AML.

***APOC1*-associated LTR2 is required for proliferation of myeloid leukaemia cells.** APOC1 has recently been shown to maintain cell survival in AML and the knockdown of *APOC1* impairs cell growth[49]. Similar findings were made in pancreatic and colorectal cancer, where *APOC1* overexpression is associated with poor prognosis[50,51]. We therefore asked whether ERV-mediated regulation of *APOC1* could affect cell growth. There is an LTR2 insertion upstream of the *APOC1* promoter (*APOC1*-LTR2, Fig. 6a, Supplementary Fig. 9A, B), which has been previously described to act as an alternative promoter in several tissues, but only accounts for up to 15% of total *APOC1* transcription[52]. In K562 and OCI-AML3 RNA-seq data, we found no evidence of *APOC1*-LTR2 promoter activity (Fig. 6a, Supplementary Fig. 8A), which we confirmed by RT-qPCR (Supplementary Fig. 9C), suggesting that *APOC1*-LTR2 could act as an enhancer element. *APOC1*-LTR2 is enriched in STAT5 and TAL1 binding and shows an increase in H3K9me3 and decrease in H3K27ac upon CRISPRi in both K562 and OCI-AML3 (Fig. 6a, Supplementary Fig. 9B). To test for a direct role of *APOC1*-LTR2 in *APOC1* gene expression and cell growth, we deleted this element in K562 cells without affecting the *APOC1* promoter (Supplementary Fig. 9A). We obtained 7 heterozygous and 8 homozygous clones from a total of 110 clones. Interestingly, none of the homozygous clones were able to grow more than 10 days in culture, suggesting that homozygous deletion may impair cell growth. To pursue the impact of *APOC1*-LTR2 on cell growth, we used lentiviral-mediated CRISPR–Cas9 delivery and performed assays in a pool of edited cells (Fig. 6b). At day 6, after GFP and puromycin selection of the two flanking sgRNAs, we observed around 60% deletion of *APOC1*-LTR2 and more than 2.5-fold reduction in *APOC1* gene expression compared with an empty vector control (Fig. 6c, d). Deletion of *APOC1*-LTR2 also led to decrease in the expression of the nearby *APOE* gene (Supplementary Fig. 9D), consistent with the results from CRISPRi (Fig. 6a). Remarkably, deletion of this element was sufficient to drive a significant suppression of cell proliferation compared with control cells (Fig. 6e). This is particularly notable given the partial nature of the deletion, emphasising the dramatic growth arrest seen in homozygous null CRISPR clones. As there is a fraction of unedited cells in the pool, we asked whether the unedited cells may outcompete edited cells over time. After day 20, the deletion was reduced to around 35%, and only 1.2-fold difference was observed in *APOC1* expression, and consequently there was no difference in cell

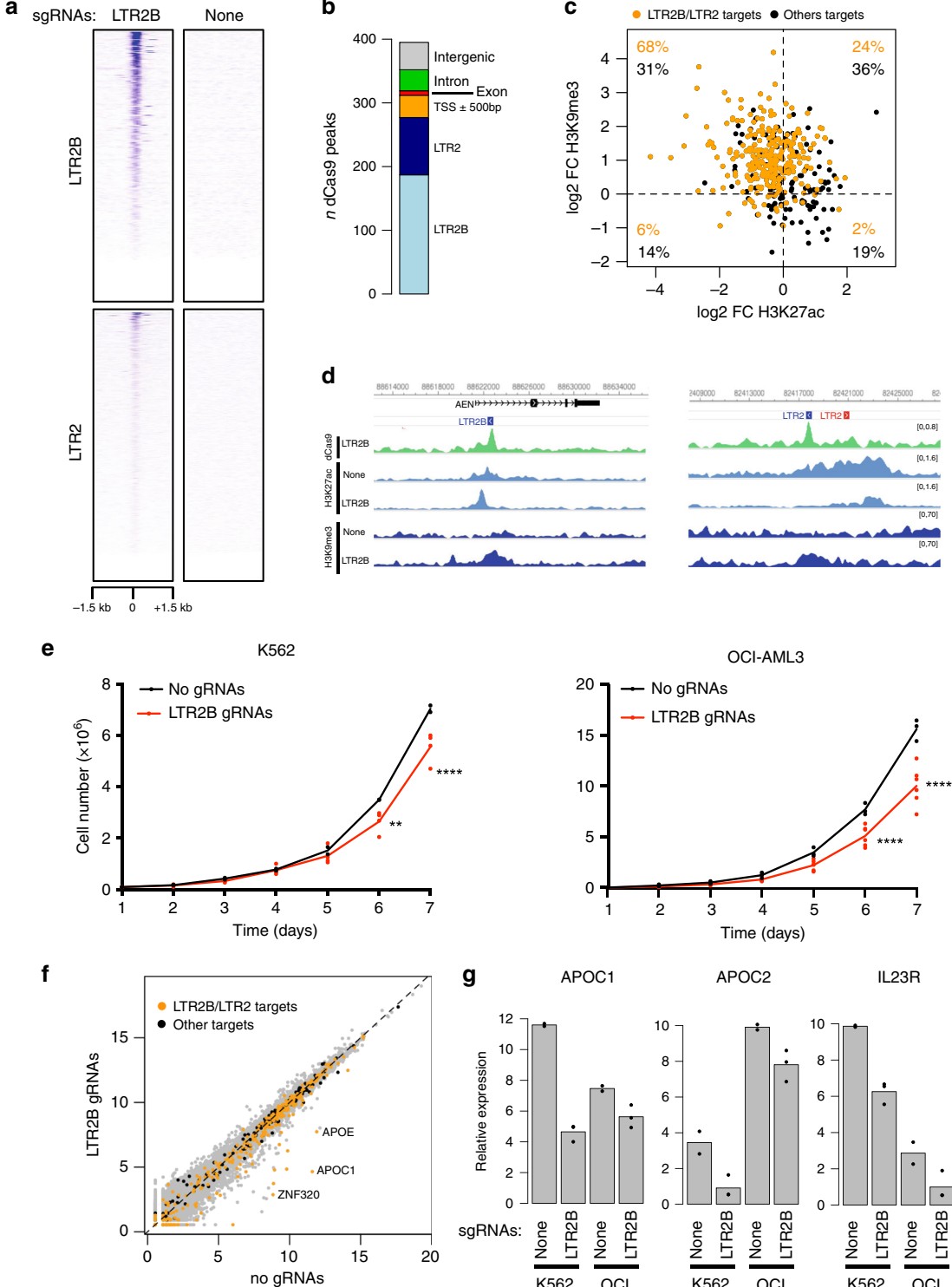

**Fig. 5 CRISPRi of LTR2B/LTR2 elements leads to impaired cell growth. a** Profile of dCas9 ChIP-seq signal over LTR2B and LTR2 elements in K562 cells expressing LTR2B sgRNAs or an empty vector ("None"). **b** Number of dCas9 peaks overlapping LTR2 and LTR2B elements, or other genomic features. **c** Log2 ratio of the ChIP-seq signal at dCas9 peaks (1 kb regions from the centre of each peak) between K562 cells expressing LTR2B sgRNAs or empty vector. Orange points highlight dCas9 peaks overlapping LTR2B or LTR2 elements. **d** Two examples of LTR2B/LTR2 elements targeted by dCas9, showing decreased H3K27ac and increased H3K9me3. **e** Cell proliferation assay in K562 (left) and OCI-AML3 (right) cells expressing LTR2B sgRNAs or an empty vector. Data are from 2 (K562) or 3 (OCI-AML3) independent assay replicates (performed at least 5 days apart) from either one (no gRNA) or two different infections (LTR2B gRNAs). **$p = 0.0096$ and ****$p < 0.0001$ (two-way ANOVA with Sidak's multiple-comparison test). Source data are provided as a Source Data file. **f** Gene expression levels in K562 cells expressing LTR2B sgRNAs or empty vector. Orange points highlight genes within 50 kb of a dCas9 peak targeting LTR2B/LTR2 elements; black points refer to genes within 50 kb of other dCas9 peaks. **g** Comparison of expression changes at selected genes between K562 and OCI-AML3 ("OCI") CRISPRi cells ($n = 3$ biological replicates).

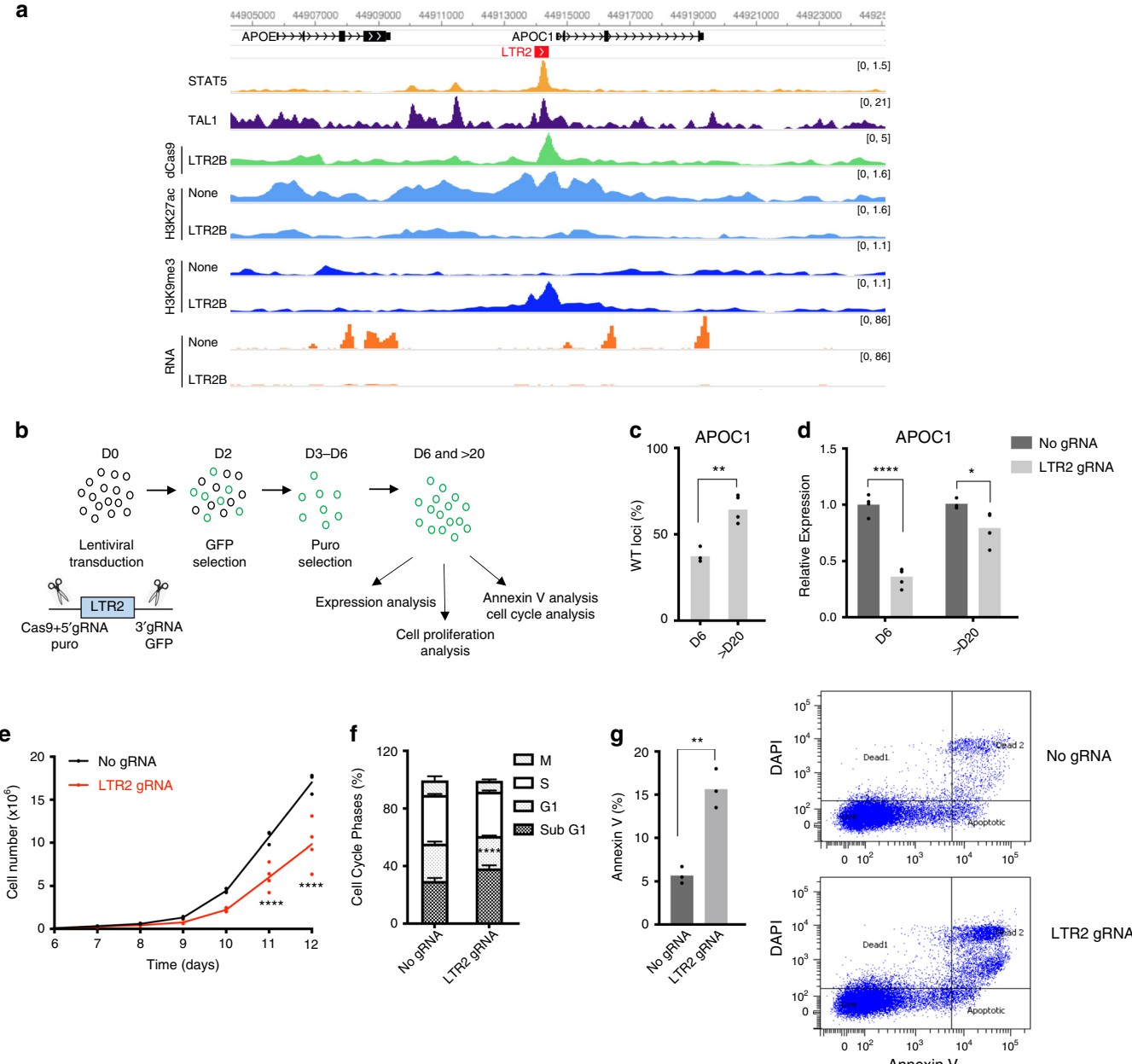

**Fig. 6 _APOC1_-LTR2 element promotes cell proliferation. a** Genome browser snapshot for _APOC1_-LTR2 element, showing TAL1, STAT5 ChIP-seq tracks for WT, H3K27ac, H3K9me3 ChIP-seq and RNA-seq tracks for no control and CRISPRi K562 cells. **b** Schematic of the experimental design to genetically excise _APOC1_-LTR2 element. **c** qPCR data from cells with _APOC1_-LTR2 excision (n = 3 (D6) and n = 4 (>D20) biological replicates, bars represent mean value; two-tailed _t_ test denotes **_p_ = 0.0036). **d** Expression data in the _APOC1_-LTR2 excision cells (n = 4 (D6) and n = 3 (>D20) biological replicates, bars represent mean value; two-way ANOVA with Tukey's multiple-comparison test, *_p_ = 0.0312 and ****_p_ < 0.0001). **e** Cell proliferation assay of control and _APOC1_-LTR2-excised cells after puromycin selection (day 6). Data are from three independent experiments, one of which used two different _APOC1_-LTR2 gRNA sets. ****_p_ < 0.0001 (two-way ANOVA with Sidak's multiple-comparison test). **f** Cell cycle profiles of control and _APOC1_-LTR2-excised cells. Data are represented as mean ± SD (n = 3 biological replicates, two-way ANOVA with Sidak's multiple-comparison test, ****_p_ < 0.0001). **g** % of Annexin V-stained cells in K562 cells upon _APOC1_-LTR2 excision (left, n = 3 biological replicates, two-tailed _t_ test, **_p_ = 0.0022). Representative flow cytometry analysis of Annexin V (right). Source data are provided as a Source Data file for (**c**–**g**).

proliferation, indicating that _APOC1_-LTR2 provides cells with a selective growth advantage (Fig. 6c, d; Supplementary Fig. 9E). To further investigate how _APOC1_-LTR2 deletion leads to impaired cell growth, we analysed cell cycle and apoptosis with flow cytometry in K562 cells at day 6. While no differences in G1, S, and G2 phases were detected, there was a significant increase in the sub-G1 population in edited cells (Fig. 6f). In agreement with this, Annexin V signal was significantly higher in edited cells compared with unedited cells at day 6 (Fig. 6g, Supplementary

Fig. 10), showing that the deletion of _APOC1_-LTR2 induces apoptosis, which is in line with known effects of _APOC1_ depletion[49–51]. As expected, this difference is much smaller after day 20 (Supplementary Fig. 9F). We also tested the effect of _APOC1_-LTR2 deletion in OCI-AML3 cells, but due to the low efficiency of Cas9 delivery and low viability of cells at day 6, we performed expression and Annexin V analysis at day 10. Similar to what we observed in K562 cells, _APOC1_-LTR2 deletion in OCI-AML3 cells led to around fourfold decrease in _APOC1_ expression and

increased Annexin V signal, and these effects were milder at day 23 (Supplementary Fig. 9G, H). Our findings indicate that the *APOC1*-LTR2 element is essential for proliferation of leukaemia cells by acting as an enhancer of the *APOC1* gene, which in turn controls cell survival via an anti-apoptotic mechanism.

Notably, DNase-seq peaks associated with *APOC1*-LTR2 in AML samples are subtler than those observed in cell lines, yet a few AML samples express relatively high levels of *APOC1* (Supplementary Fig. 11A, B). Interestingly, overall survival curves based on TCGA data suggest that a small proportion of patients with high *APOC1* expression have a poorer prognosis, a pattern that is also seen in patients with high *APOE* expression (Supplementary Fig. 11C, D).

## Discussion

Here, we demonstrate that particular ERVs are used as regulatory elements to activate gene expression in AML, which may be exploited by cancer cells to help drive disease phenotypes and cancer progression. Many of these ERVs are also active in CD34+ progenitor cells and are therefore not cancer-specific, but they may nonetheless be used to support a gene expression programme that blocks cellular differentiation, a key hallmark of AML. Genetic and epigenetic perturbation experiments, such as the ones presented here, allow us to distinguish between ERVs that support oncogenesis and those whose activation is secondary to cellular dedifferentiation.

It had been previously postulated that the epigenetically relaxed state of cancer cells provides a window of opportunity for ERV activation, triggering their intrinsic regulatory capacity[9,24,53]. However, to the best of our knowledge, all examples to date supporting this hypothesis have involved activation of cryptic promoters to drive expression of adjacent genes[24,27]. Whilst we uncovered some examples of chimeric transcripts starting from ERVs in AML (e.g., LTR2C-*SAGE1* and LTR2B-*RHEX*), which are not present in differentiated myeloid cells, our analyses suggest that active A-DARs mainly harbour chromatin signatures of enhancers.

We identified multiple ERV elements with strong evidence supporting their role as bona fide gene regulators: (1) we found striking correlations between differential chromatin accessibility at 20 ERVs and the expression of nearby genes, some of which have been linked to AML prognosis (Fig. 1e, f), (2) CRISPR-mediated genetic editing experiments revealed an additional 5 ERVs that act as enhancers in leukaemia cells (Fig. 4, Supplementary Fig. 8E, Fig. 6) and (3) CRISPRi identified another 13 different elements whose epigenetic silencing led to the down-regulation of nearby genes (Supplementary Data 7). A more exhaustive search would likely have revealed additional regulatory elements, namely via epigenetic silencing of other ERV families. Moreover, given the heterogeneity of the disease, inclusion of additional primary AML data or a focus on specific AML subtypes may have uncovered other ERV families/loci of interest.

Despite the growing evidence that ERVs can act as regulatory elements in different cancers, there are limited examples for their inappropriate activation contributing to oncogenesis, a term coined as onco-exaptation[54]. The term has been frequently used to describe the gain of regulatory activity at TEs in cancer. Our view is that, similar to the term exaptation[55], onco-exaptation requires that this new regulatory activity provides the cancer cell with a selective advantage. Strong demonstrations of such adaptive roles are scarce. Notably, the Wang lab recently showed that an AluJb element acts as an oncogenic promoter to drive *LIN28B* expression and tumour progression in lung cancer[27]. In our study, we identified an LTR2 element, the genetic and epigenetic

perturbation of which suppressed cell growth and induced apoptosis of leukaemia cell lines by altering lipid-related *APOC1* expression. Despite the striking cellular phenotype in cell lines, comprehensive analyses of primary AML samples are warranted to demonstrate whether these regulatory ERVs are sufficient to provide survival advantages for cancer cells in vivo and contribute to prognosis of AML. Notably, we observed that AML patients with high *APOC1* or *APOE* expression demonstrate significantly lower overall survival rate. A considerably larger number of patients would be necessary to confirm this finding, although independent datasets have led to similar observations in color-ectal and pancreatic cancer[50,51]. APOC1 is also activated in monocyte-to-macrophage differentiation[56], raising the possibility that *APOC1*-LTR2 may play other roles in haematopoiesis outside of AML.

Given their repetitive nature, one intriguing question is why particular ERVs within a family are recurrently activated in AML to drive nearby gene expression, yet the majority of them are functionally neutral. One explanation lies in the nature of inter- and intra-cellular epigenetic heterogeneity that increases during malignancy formation. This gives rise to epigenetic activation of a set of ERVs, as proposed in the epigenetic evolution model[24]. Accordingly, cells harbouring activated ERVs that drive onco-genes gain a selective advantage and increase in frequency during cancer evolution. Therefore, clonal expansion of these cells will enable the detection of oncogenic ERVs in a cell population. However, whether ERV activation contributes to cancer evolution or is simply a consequence of the molecular state of cancer remains a matter of debate.

Irrespective of whether epigenetic heterogeneity at ERVs con-tributes to tumour evolution, distinct patterns of ERV activity are observed across different AML patients (Supplementary Fig. 1A). These differences appear to be partly driven by the underlying mutational profiles. We also identified a SNP within an ERV that seemingly affects its regulatory activity by altering a TF-binding site (Supplementary Fig. 7C), suggesting that genetic variation within ERVs also contributes to inter-individual differences in ERV activity. Finally, younger ERVs such as LTR5_Hs are structurally polymorphic within the human population[12,57], adding another layer of genetic variation. Regulatory ERVs may therefore foster genetic, epigenetic and transcriptional hetero-geneity of the disease with potential to contribute to clinical outcomes. One significant consequence of the molecular hetero-geneity of AML is the escape of resistant clones from treatment, resulting in high relapse rates. It will be therefore interesting to discover to which extent the ERV-derived heterogeneity con-tributes to inter-individual differences in response to AML therapies.

Our work reveals ERVs as potentially oncogenic enhancers in AML. These data highlight the significance of expanding the search for oncogene drivers to the repetitive part of the genome, which may pave the way for the development of novel prognostic and therapeutic approaches.

## Methods

**Cell culture and cell proliferation assays**. OCI-AML3, MOLM-13 and HL-60 cell lines were provided by Professor Brian Huntly, originally sourced from DSMZ; K562 was provided by Dr. Farideh Miraki-Moud, originally derived by Lozzio & Lozzio[58]; 293T cells were provided by Dr. Ana O'Loghlen, originally derived by DuBridge et al.[59]. 293T cells and human leukaemia cell lines K562, OCI-AML3, MOLM-13 and HL-60 were routinely cultured in RPMI 1640 (and DMEM (HEK293T)) supplemented with 10% foetal bovine serum, 2 mM glutamax and 1% penicillin/streptomycin at 37 °C in 5% carbon dioxide. Cells were maintained and split every 2–3 days.

For cell proliferation assays, exponentially growing cells were plated in 24-well plates ($1 \times 10^5$ cells/ml). Every 2–3 days, media were replaced, and cells were split into $1 \times 10^5$ cells/ml. The viable cells were counted daily for 6 days.

**Cell cycle and apoptosis assays**. Cell cycle assay was performed using muse cell cycle kit by following the manufacturer's instructions (Millipore), and the cells were analysed by BD FACS Canto II. For apoptosis assay, the cells were stained by an annexin V 647 (Thermofisher Scientific) and DAPI and analysed by BD FACS Canto II.

**CRISPR–Cas9-mediated LTR disruption**. For CRISPR/Cas9 deletion of LTRs, sgRNA oligonucleotides (Sigma-Aldrich) targeting upstream and downstream of LTRs of interest were annealed and cloned into modified eSpCas9 (1.1) vector (Addgene 71814, deposited by Feng Zhang), which expresses GFP. K562 cells were nucleofected with eSpCas9 plasmid containing gRNAs using amaxa nucleofector kit V. Two days later, cells expressing GFP were sorted on a FACS Aria II, and single cells were plated onto a 96-well plate. After 2 weeks, cells were genotyped by PCR, and the gene expression of LTR-knockout cells was analysed by RT-qPCR.

For LTR2-APOC1 deletion, 5′ sgRNAs (Sigma-Aldrich) were cloned into lentiCRISPR v2 (Addgene 52961) and 3′ sgRNAs were cloned into lenti_sgRNA_EFS_GFP (Addgene 65656) vector. OCI-AML3 and K562 cells were transduced with the lentiviral vectors containing sgRNAs and selected for GFP and puro. % of WT loci was determined by qPCR using APOC_R and APOC_I genotyping primers listed in Supplementary Data 8. The cells were cultured around 3 weeks for RNA expression and phenotypical analysis.

**CRISPRi-mediated silencing of LTRs**. sgRNAs (Sigma-Aldrich) targeting multiple LTR copies were cloned into lentiviral expression vector pKLV-U6gRNA(BbsI)-PGKpuro2ABFP (Addgene 50946, deposited by K. Yusa). For LTR silencing, OCI-AML3 and K562 cells were first transduced with the lentiviral vector pHR-SFFV-KRAB-dCas9-P2A-mCherry (Addgene 60954, deposited by Jonathan Weissman), sorted for mCherry on a FACSAria II. Cells expressing mCherry were then subsequently transduced with the lentiviral sgRNA expression vector. Two days later, the cells expressing both mCherry and BFP were sorted and cultured for transcriptional and chromatin analyses.

**Lentiviral production and transduction**. Lentivirus was produced in 293T cells by triple transfection with delivery vector and the packaging plasmids psPAX2 and pMD.G. The viral supernatants were collected 48 h after transfection and filtered through a 0.45 µM filter. Target cells were transduced with lentiviral supernatant supplemented with 4 µg/mL polybrene.

**RNA isolation and RT-qPCR**. RNA was extracted using AllPrep DNA/RNA mini kit (Qiagen 80204) and DNAse treated with the TURBO DNA-free™ Kit (Ambion, AM1907). RNA (1 µg) was retrotranscribed using Revertaid Reverse Transcriptase (Thermo Scientific EP0441), and the cDNA was diluted 1/10 for qPCRs using MESA BLUE MasterMix (Eurogenentec, 10-SY2X-03+NRWOUB) on a Light-Cycler® 480 Instrument II (Roche). A list of primers used can be found in Supplementary Data 8.

**RNA-seq library preparation**. Ribosomal RNA-depleted RNA-seq libraries were prepared from 200 to 500 ng of total RNA using the low-input ScriptSeq Complete Gold Kit (Epicentre). Libraries were sequenced on an Illumina NextSeq 500 with single-end 75-bp reads.

**Chromatin immunoprecipitation**. Approximately, $10^7$ cells were fixed with 1% formaldehyde for 12 min in PBS and quenched with glycine. Chromatin was sonicated using a Bioruptor Pico (Diagenode), to an average size of 200–700 bp. Immunoprecipitation was performed using 75 µg of chromatin and 5 µg of Cas9 antibody (Diagenode #C15200229-100) or 15 µg of chromatin and 2.5 µg of H3K27ac and H3K9me3 antibody (Active Motif #3913, Diagenode #C15410193). The final DNA purification was performed using the GeneJET PCR Purification Kit (Thermo Scientific #K0701), and DNA was eluted in 80 µL of elution buffer. This was diluted 1/10 and analysed by qPCR, using the KAPA SYBR® FAST Roche LightCycler® 480 2× qPCR Master Mix (Kapa Biosystems, Cat. KK4611). A list of primers used can be found in Supplementary Data 8.

**Library preparation and sequencing for ChIP-seq and DNase-seq**. ChIP-seq and DNase-seq libraries were prepared from 1 to 5 ng of ChIP DNA or DNase DNA samples using NEBNext Ultra II DNA library Prep Kit (Illumina). Libraries were sequenced on an Illumina NextSeq 500 with single-end or paired-end 75-bp reads.

**Chromatin accessibility assay**. To assess chromatin accessibility, 5 million cells were resuspended in RSB buffer (10 mM NaCl, 3 mM $MgCl_2$ and 10 mM Tris-Cl, pH 7.4). After cell lysis, the nuclei were digested with DNase I with 0, 0.1, 2, 5, 15 and 30 U for 10 min at 37 °C. Digests were inactivated by the addition of 50 mM EDTA. RNA and proteins were digested by RNase A (0.5 mg/ml) for 15 min at 37 °C and then by proteinase K (0.5 mg/ml) for 1 h at 65 °C. DNA was purified by phenol–chloroform extraction and ethanol precipitation. The resuspended DNA was analysed by qPCR, using the KAPA SYBR® FAST Roche LightCycler®

480 2× qPCR Master Mix (Kapa Biosystems, Cat. KK4611), and chromatin digested with 15 U was selected for library preparation and sequencing.

**Primary processing of high-throughput sequencing data**. Reads from high-throughput sequencing data generated here or from external datasets (Supplementary Data 9) were trimmed using using Trim Galore. ChIP-seq and DNase-seq data were aligned to the hg38 genome assembly using Bowtie2 v2.1.0[60], followed by filtering of uniquely mapped reads with a custom script. ChIP-seq peak detection was performed using MACS2 v2.1.1[61] with -q 0.05; for histone marks the option --broad was used. DNase-seq peak detection was performed using F-seq v1.84[62] with options -f 0 -t 6. RNA-seq data were mapped using Hisat2 v2.0.5[63] with option --no-softclip. Raw read counts for each gene were generated in Seqmonk with the RNA-seq quantitation pipeline, and normalised gene expression values calculated with the variance-stabilising transformation in DESeq2[64]. BigWig tracks were generated using the bamCoverage function of deepTools2.0, with CPM normalisation and 200-bp bin size. Other processed data from Blueprint, ENCODE and other sources (Supplementary Data 9) were downloaded as peak annotations or expression values (e.g., FPKM).

**DHS enrichment at repeat families**. DHSs (i.e., DNase-seq peaks) were intersected with the Repeatmasker annotation, and the number of overlapped DHSs per repeat family calculated. For comparison, 1000 random controls were generated by shuffling the DHSs in a given sample, avoiding unmappable regions of the genome. $p$ Values were calculated based on the number of random controls for which the number of DHS overlaps displayed more extreme values (at either tail of the distribution) than those seen with the real DHSs. Enrichment values were calculated by dividing the number of real DHS overlaps with the mean number of DHS overlaps in the random controls. Significantly enriched repeat families had (1) $p <$ 0.05, (2) >2-fold enrichment,and (3) >20 copies overlapped by DHSs. Selected families were significantly enriched for DHSs in at least one of the cell lines analysed (HL-60, OCI-AML3 and MOLM-13) and in >10% of AML samples.

**Mutational profile analysis**. A-DAR elements overlapping DHSs in at least one sample were selected, and a correlation matrix built based on the patterns of DHS overlap between samples. These were compared with the AML mutational profiles extracted from the respective publications[5,6]. Correlation coefficients between AML samples sharing a particular mutation were compared with correlation coefficients between samples without the mutation.

**Identification of active A-DAR promoters**. Aligned BAM files from Blueprint RNA-seq data were processed using StringTie v1.3.3b[65] with options --rf -G to generate sample-specific transcriptome assemblies guided by the GENCODE annotation v26. Spliced transcripts initiating at A-DAR elements were then identified by intersecting the TSSs of multi-exon transcripts of A-DAR annotations. A-DAR elements with TSSs in AML samples but not in differentiated cells were selected, and the associated transcripts visually inspected to identify those with evidence of splicing into GENCODE-annotated genes. TSSs were also checked against the FANTOM5 robust CAGE peak set (hg38 version, with fairly remapped and newly identified peaks).

**K562 TF ChIP-seq analysis**. ENCODE TF ChIP-seq peak files from K562 (Supplementary Data 9) were downloaded and intersected with A-DAR annotations, as well as with a randomly shuffled version of these elements. TFs significantly enriched (corrected $p < 0.05$) in at least one of the A-DAR families, covering at least 5% of the elements in that family, were selected. For each TF, average enrichment values were calculated across technical and biological replicates, as well as independent ChIP-seq experiments of the same TF.

**TF motif analysis**. Motif analysis of A-DARs was performed using the AME and FIMO tools of the MEME SUITE v5.0.1[66] using the HOCOMOCO v11 human TF motif database. Motifs enriched in at least one A-DAR family were identified using AME, and motif frequency and location extracted using FIMO. Consensus sequences were downloaded from Dfam[31].

**CRISPRi ChIP-seq and RNA-seq analyses**. Normalised H3K27ac and H3K9me3 ChIP-seq read counts were extracted around dCas9 peaks (±500 bp from the peak centre). Genes within 50 kb of a dCas9 peak were considered as putative direct targets of CRISPRi. Differential gene expression analysis was performed using DEseq2[64].

**Reporting summary**. Further information on research design is available in the Nature Research Reporting Summary linked to this article.

## Data availability

High-throughput sequencing data that support the findings of this study have been deposited in the Gene Expression Omnibus (GEO) with the accession code GSE136764. A list of publicly available datasets used in this study are listed in Supplementary Data 9.

In addition, the following public databases were used: GENCODE v26 [https://www.gencodegenes.org/human/release_26.html], FANTOM5 [https://fantom.gsc.riken.jp], Dfam [https://dfam.org/home] and HOCOMOCO v11 [https://hocomoco11.autosome.ru]. Other data that support this study are available from the corresponding author upon reasonable request. Source data are provided with this paper.

## Code availability

Scripts used for data analysis are available from GitHub https://github.com/MBrancoLab/Deniz_2019_AML.

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

## Acknowledgements

We thank Yasmine Benbrahim for ideas informing bioinformatic analyses, the Dawson lab for their guidance in lentiviral transduction of AML cell lines, Brian Huntly for providing OCI-AML3, MOLM-13 and HL-60 cell lines, Gary Warnes for flow cytometry analysis and Diego Villar and Jenny Frost for critical reading of the paper. This work was supported by funding from Barts Charity (Small Project Grants—MGU0462). O.D. received funding from the People Programme (Marie Curie Actions) of the European Union's Seventh Framework Programme (FP7/2007–2013) under REA grant agreement no. 608765. M.R.B. was supported by a Sir Henry Dale Fellowship (101225/Z/13/Z), jointly funded by the Wellcome Trust and the Royal Society. This study makes use of data generated by the Blueprint Consortium. A full list of the investigators who contributed to the generation of the data is available from www.blueprint-epigenome.eu. Funding for the project was provided by the European Union's Seventh Framework Programme (FP7/2007–2013) under grant agreement No. 282510—BLUEPRINT. This research utilised Queen Mary's Apocrita HPC facility, supported by QMUL Research-IT[67].

## Author contributions

O.D. and M.R.B. designed the study and experiments and wrote the paper. O.D. performed cell culture, DNase-seq, ChIP-seq, RNA-seq, CRISPR, CRISPRi and cellular phenotyping. M.A. generated the ZNF611-LTR2B KO. C.D.T. assisted in the design and execution of CRISPR experiments. A.R.M. performed the overall survival analyses. M.A.D. assisted in the establishment of CRISPRi cell lines. M.R.B. performed the bioinformatic analyses.

## Competing interests

The authors declare no competing interests.
