## [Peer Review File · Nature Communications]

Reviewers' comments:

Reviewer #1 (Remarks to the Author):

In this manuscript by Deniz and co-workers the authors investigate if endogenous retroviruses serve as a source of enhancers in AML and how this may influence tumor heterogeneity and evolution. Overall, this is a well-written manuscript that is easy to understand and the message is clear.

The study appears to be well designed and the results are nicely presented and interpreted. The choice of methodology is clearly state-of-the-art. My main concern with the manuscript is if these findings are relevant to AML or if they primarily represent *in vitro* artefacts found in K562-cells? This concern somewhat dampens my enthusiasm with the manuscript and puts into question the potential impact of this study.

Major comments

1. It would be very valuable if the authors could strengthen the AML-dataset to make their observations less preliminary. Given the high heterogeneity of the AML-samples it is obvious that the study is underpowered. At least the authors should discuss this limitation more clearly.

2. The title is overselling the results and should be toned down. The evidence that ERVs are oncogenic enhancers is rather weak.

3. The CRISPRi experiment is interesting – but also difficult to fully interpret. Additional control experiments would be very valuable. For example, did the authors try the same experimental setup with a gRNA targeting a different ERV family of similar size that do not carry enhancer marks? It would also be very interesting to extend this experiment to a more relevant cell type, such as primary CD34+ cells.

4. The CRISPR experiment in Fig 6 is also very interesting. However, additional controls would be very valuable. For example, what happens if they cut out a similar size piece of DNA just upstream or downstream of the ERV?

In summary, I found this to be a very robust study that uses state-of-the-art technologies to study ERV biology. My only problem with the study is the weak link between the observations done in K562 cells and actual AML samples.

Reviewer #2 (Remarks to the Author):

Endogenous retroviral elements (ERVs), particularly their LTRs, have natural transcriptional regulatory capacity. While most of the several hundred thousand LTRs in the human genome are epigenetically repressed, some are active in certain normal contexts, providing promoters and enhancers to host genes. There has been increasing interest in determining if “activation” of normally repressed LTRs in cancer could lead to deregulation of genes and contribute to oncogenesis. As the authors point out, previous efforts have mainly focused on LTR promoter activity in cancer for mainly practical reasons. This study examines the role of ERV LTRs, particularly as enhancers, in AML, which is a heterogeneous disease with generally poor outcomes. They first search for potential enhancers in ERVs or other repeats by intersecting DNase hypersensitive sites (DHSs) in 32 AML samples (as generated by the Blueprint epigenome project) with Repeatmasker annotations. As normal controls they use data from CD34+ cells, macrophages and monocytes. They identify six families of ERV LTRs that are enriched for DHSs in some AML

samples from two datasets and in a few AML cell lines compared to normal controls and call these A-DARs (for AML DHS-associated repeats). These A-DARs are associated with histone marks typical of enhancers and expression of nearby genes correlates with A-DAR chromatin status. They also looked at TF ChIP-seq data from the AML cell line K562 and identified over 200 TFs that bind at least 5% of LTRs of a given family, some of which are involved in hematopoiesis and AML. TF motif analysis revealed that, in most cases, presence or absence of the motif does not correlate with DHS status of a particular element. They then proceed with more functional experiments and show that deleting 3 select LTRs in AML cell lines with CRISPR leads to reduction of nearby gene expression. CRISPRi experiments targeting the LTR2 and 2B families in two AML cell lines showed some genes are downregulated and that cell proliferation was reduced in both cell lines. Finally, they focused on an LTR2 upstream of the APOC1 gene since APOC1 has previously been shown to maintain cell survival in AML. Deleting this LTR leads to a decrease in APOC1 expression and significant decreases in cell proliferation as judged by lack of survival of homozygous edited clones and increases in apoptosis. Examination of survival data in TCGA shows that high APOC1 expression is correlated with poorer prognosis.

Overall, this is a very well done study and the logic of the paper flows nicely. The methods are generally well described. It is a comprehensive body of work and adds significantly to the growing literature on the potential roles of ERVs in cancer. I just have some relatively minor concerns:

1. This is not a concern but rather an complement. CRISPR deletion experiments showed that a particular LTR5B element acts as an enhancer of the nearby RPL7L1 gene. Instead of stopping there, the authors noted that this LTR has a SNP where the minor allele disrupts a MEK binding site motif. Analysis of GTEx data showed that the minor allele is associated with lower RPL7L1 expression in blood. While a minor point in the paper, it demonstrated to the field that investigators should consider sequence variation within LTRs, not just the presence of the LTR, when evaluating their regulatory effects – an aspect which has been greatly understudied to date.

2. Please add a statement to confirm that the editing sites of the non-viable APOC1-LTR2 homozygous clones were contained within the LTR2 and did not extend into the flanking sequences. Admittedly, it is unlikely that this happened but it would strengthen the argument.

3. One limitation of a study like this that the authors should at least discuss is the common problem of declaring something “cancer-specific” when comparing data from a large set of cancer samples to data from a much smaller number of normal samples. Since many of their analyses require only that the epigenetic mark or promoter activity, etc. be present in at least two AML samples but not in the normal differentiated cells (monocytes and macrophages), how can the authors be confident that a larger collection of normal cell samples might also have a few which display that particular mark or activity? A related problem is determining which “normal” cells to compare to AML samples. Here the authors use data from monocytes and macrophages and one could argue that fully differentiated cells are not the most appropriate control to compare to a poorly differentiated blood cancer. Perhaps some of the differences in ERV epigenetic marks or promoter activity have nothing to do with the cancer but just to the degree of differentiation. In other words, one wonders what fraction represent true “activation” of an LTR which is normally never active versus cases where the LTR is active in some normal tissues or developmental stages but just not in the few normal samples that the authors used to compare to the AMLs. The authors should add some discussion acknowledging/addressing these issues.

4. In the first figure (fig 1B) and associated text, five transposon families appear to be enriched for DHSs in all samples and were therefore eliminated from further analysis. Two of these (TAR1 and MSR1) are satellite repeats and U1 is a snoRNA. These are not transposons and should not have been included in the screening method in the first place. A Blat search shows that most copies of the MER57E3 LTR are in the first intron of ZNF genes, having amplified as part of ZNF gene

duplications, so also is not the typical dispersed transposon and should probably also not even be shown. Its DHS enrichment could well be due to proximity to ZNF gene promoters. The last one, LTR13, is also somewhat strange as many copies seem to be located in regions that don't even align with Rhesus (according to a brief search using BLAT).

5. For the data in Figure 1E, which shows the upregulated genes associated with a DHS at a nearby ERV, please provide a supplementary table with the gene names, coordinates of the ERV and the distance of the ERV to the TSS of the gene.
6. Related to the above data, the authors discuss three genes at the bottom of page 6 that have some link to AML prognosis. Please elaborate a little more. Is up or down regulation of these genes linked to worse outcome?
7. In the section where the authors look for LTRs acting as promoters (page 7), while "promoter effects" seem less important than enhancer effects, At least one gene (RHEX) in the associated Table is probably worth mentioning. LTR-RHEX transcripts are present in a large fraction of AMLs and the LTR2B is one of two annotated promoters in RefSeq. Since this gene is "regulator of hemoglobinization and erythroid cell expansion", it would be worth commenting on it since it shows up in such a large fraction of AMLs.
8. Related to the above, please include a paragraph on how ab initio transcriptome assembly was performed. As well, it is stated in the methods that the assembled transcripts were "visually inspected". Were these checked against other sources (eg. ESTs or Gencode or FANTOM5) for evidence that the LTRs are indeed the promoters? As an example, for the SUOX gene, the LTR12C seems to provide part of an internal exon in UCSC and is certainly not the promoter.
9. Several of the quoted references are not in the reference list. Please fix.

Reviewed by Dixie Mager

Reviewer #3 (Remarks to the Author):

Deniz et al have mined publicly available data from AMLs, leukemic cell lines and monocytes/macrophages and investigated the frequency and identity of transposable elements at sites of chromatin accessibility, epigenetic marks of active/inactive promoters/enhancers to focus on a limited number of families to investigate their gene regulatory potential. As acknowledge, the concept is not novel, but this is an excellent example of using public data to explore a hypothesis and performing a series of valid experiments to elegantly prosecute their case. My comments are as follows.

Figure 1D- were the expression levels averaged across BluePRINT AMLs? This wasn't clear to me.

Figure 2C- the presence/absence of a particular histone mark at a particular site was called using MACS2 with set thresholds. How were these normalised across samples?

Figure 2F- Indicate more clearly in the figure that the profiles are of H3K27Ac and given the occasionally noisy profiles- mark the peaks that were called by MACS

Figure 3E- Given the visual association that motifs were enriched at DHS elements, which the authors discount as insignificant- this should be backed up by valid statistics. There is publicly available Hi-C data in K562 cells. It should be possible to directly link the TF bound TEs with gene promoters.

Figure 4B- Following CRISPR of elements in k562, gene expression values were shown for three ZNF genes. The displayed window in 4A only shows one presumably for practical considerations but it leaves the reader wondering whether RPL7L1 and BIK exist in gene deserts or selective data representation. Use of K562 Hi-C data could potentially show the TE/promoter links. Show 4B as in

5G so the number of clones analysed is evident.

Figure 5- deposition of dCas9 at intronic regions can disrupt transcript elongation independently of disrupting a regulatory element. What proportion of genes with intronic dCas9 (5B) showed reduced expression and could this have impacted on cell growth?

Figure 6- I would be interested to know what the chromatic accessibility/histone profiles at LTR2 adjacent to APOC1 look like in CD34+?

General-

There were a number of references in text that have not transferred to the reference list. This should be corrected.

TF binding data is limited to K562 cells. There is publicly available TF data in CD34+ (BloodChIP)

Have the authors looked for the presence of TEs at heptad binding sites?

When focussing on one or other family of LTR from one figure to the next, the authors should justify the selection

Legend 1- line 714- incomplete credit. Line 726- add colours against expression (orange), DHS peak (blue)

Reviewer #1

The study appears to be well designed and the results are nicely presented and interpreted. The choice of methodology is clearly state-of-the-art. My main concern with the manuscript is if these findings to are relevant to AML or if they primarily represent *in vitro* artefacts found in K562-cells? This concern somewhat dampens my enthusiasm with the manuscript and puts into question the potential impact of this study.

We appreciate the reviewer's concern and agree that the functional outcomes of deleting or silencing ERVs require a follow-up study in primary AML patient samples. We are currently building the expertise to perform such experiments, but they are technically very challenging and constitute a significant endeavour that goes beyond the scope of the present paper. Here we identified promising candidate ERV families and loci that will guide those experiments in primary samples. Importantly, the choice of candidates was strongly based on epigenomic and transcriptomic data from primary AML samples:

- a) We analysed two independent AML cohorts, which in total have 64 patient samples. The results of our DNase-seq analysis were consistent between these two cohorts.
- b) In our genetic deletion experiments we only selected candidate ERVs that overlap with DHSs in multiple AML samples and whose associated genes are highly expressed in AML.
- c) Our CRISPR-based epigenetic editing experiments were focused on an ERV family that is more active in AML than in CD34+ cells. These experiments were carried out in two leukaemia cell lines, with consistent results.

In reply to a point raised by reviewer #3, we also now include an analysis of transcription factor ChIP-seq data from primary CD34+ cells. Furthermore, we have adjusted the title of the paper (see below) and added the following text to the discussion: "Despite the striking cellular phenotype in cell lines, comprehensive analyses of primary AML samples are warranted to demonstrate whether these regulatory ERVs are sufficient to provide survival advantages for cancer cells *in vivo* and contribute to prognosis of AML".

1. It would be very valuable if the authors could strengthen the AML-dataset to make their observations less preliminary. Given the high heterogeneity of the AML-samples it is obvious that the study is underpowered. At least the authors should discuss this limitation more clearly.

We analysed two of the largest existing DNase-seq datasets in AML, totalling 64 patient samples. We have also analysed the most comprehensive epigenomic profiling of AML to date (H3K27ac, H3K4me1, H3K4me3 and H3K9me3) from the Blueprint consortium (32 samples). In our opinion this would not classify as a preliminary analysis. Whilst we agree that heterogeneity affects the detection of active ERVs, our analysis clearly had sufficient power to detect several relevant ERV families and individual loci that are active across multiple samples. Future work will focus on specific AML subtypes. We have added the following to the discussion: "...given the heterogeneity of the disease, inclusion of additional primary AML data or a focus on specific AML subtypes may have uncovered other ERV families/loci of interest."

2. The title is overselling the results and should be toned down. The evidence that ERVs are oncogenic enhancers is rather weak.

We agree with the reviewer and have changed the title to "Endogenous retroviruses are a source of enhancers with oncogenic potential in acute myeloid leukaemia".

3. The CRISPRi experiment is interesting – but also difficult to fully interpret. Additional control experiments would be very valuable. For example, did the authors try the same experimental setup with a gRNA targeting a different ERV family of similar size that do not carry enhancer marks? It would also be very interesting to extend this experiment to a more relevant cell type, such as primary CD34+ cells.

We agree with the reviewer that the CRISPRi experiment has some caveats, as it may suffer from indirect and off-target effects. Nevertheless, it is a powerful way to screen a whole ERV family for loci of interest. These experiments should be followed up by genetic deletion experiments to validate candidate loci, which is what we have done here for LTR2-APOC1. We were also very stringent when selecting ERVs that likely directly control nearby genes, and controlled for off-target effects.

The suggested experiment of silencing a 'neutral' ERV family, whilst interesting, also has its caveats. Namely, it is possible that a particular element in a family is phenotypically important, even if the family on average does not look to be particularly active from an enhancer perspective. Performing CRISPRi experiments in primary cells is a valid point to show whether ERVs are phenotypically important *in vivo*, but as argued above, these are very challenging experiments and beyond the scope of this paper.

4. The CRISPR experiment in Fig 6 is also very interesting. However, additional controls would be very valuable. For example, what happens if they cut out a similar size piece of DNA just upstream or downstream of the ERV?

This is an interesting point that would question whether LTR2 acts as an enhancer or a neutral fragment that affects gene expression by an unknown mechanism. We believe that additional experiments along these lines would address something much more fundamental about gene regulation that no other study (in any context) has addressed. Such a study would merit a lot more attention than a single locus, and many more experiments. Importantly, our main claim remains valid that the LTR2 insertion controls APOC1 expression and supports cell proliferation. Its role as an enhancer is strongly suggested by the epigenomic data.

Reviewer #2

Overall, this is a very well done study and the logic of the paper flows nicely. The methods are generally well described. It is a comprehensive body of work and adds significantly to the growing literature on the potential roles of ERVs in cancer. I just have some relatively minor concerns:

1. This is not a concern but rather an complement. CRISPR deletion experiments showed that a particular LTR5B element acts as an enhancer of the nearby RPL7L1 gene. Instead of stopping there, the authors noted that this LTR has a SNP where the minor allele disrupts a MEFK binding site motif. Analysis of GTEX data showed that the minor allele is associated with lower RPL7L1 expression in blood. While a minor point in the paper, it demonstrated to the field that investigators should consider sequence variation within LTRs, not just the presence of the LTR, when evaluating their regulatory effects – an aspect which has been greatly understudied to date.

We thank the reviewer for kind comments.

2. Please add a statement to confirm that the editing sites of the non-viable APOC1-LTR2 homozygous clones were contained within the LTR2 and did not extend into the flanking sequences. Admittedly, it is unlikely that this happened but it would strengthen the argument.

The editing sites had to be designed outside the LTR2 element due to the sequence similarities of copies within LTR2 family. As shown below, we have chosen the gRNAs as close to LTR2 element borders as possible not to delete extra sequences. We have added this schematic to Supplementary Figure 9.

3. One limitation of a study like this that the authors should at least discuss is the common problem of declaring something “cancer-specific” when comparing data from a large set of cancer samples to data from a much smaller number of normal samples. Since many of their analyses require only that the epigenetic mark or promoter activity, etc. be present in at least two AML samples but not in the normal differentiated cells (monocytes and macrophages), how can the authors be confident that a larger collection of normal cell samples might also have a few which display that particular mark or activity? A related problem is determining which “normal” cells to compare to AML samples. Here the authors use data from monocytes and macrophages and one could argue that fully differentiated cells are not the most appropriate control to compare to a poorly differentiated blood cancer. Perhaps some of the differences in ERV epigenetic marks or promoter activity have nothing to do with the cancer but just to the degree of differentiation. In other words, one wonders what fraction represent true “activation” of an LTR which is normally never active versus cases where the LTR is active in some normal tissues or developmental stages but just not in the few normal samples that the authors used to compare to the AMLs. The authors should add some discussion acknowledging/addressing these issues.

We appreciate the reviewer’s concern regarding the seemingly unbalanced number of AML vs. differentiated samples. However, whilst we were limited by the number of monocyte/macrophage samples in the DNase-seq analysis, the RNA-seq and ChIP-seq analyses include a higher number of these samples, giving us confidence that our results are robust (especially since differentiated cell types are more homogeneous than AML samples). ChIP-seq data were from 15 samples from differentiated cells (7 macrophage and 8 monocyte) and RNA-seq from 14 samples of differentiated cells (6 macrophage and 8 monocyte). We have now indicated in the text (lines 164/165 and 211/212).

Regarding the second point, we agree with the reviewer that ERV activation may be associated with a less differentiated cell state, and not all ERVs will be specific to AML. Indeed, we found that 5 of our 6 ERV families of interest are also active in CD34+ cells, but we argued that ERVs associated with a stem cell state “may be exploited by cancer cells to promote cell proliferation and survival” (line 130). Our goal was to identify any ERVs that are putatively driving oncogenesis, and this includes regulatory elements that impair cellular differentiation. AML is essentially a block in the differentiation of myeloid progenitors, which is why the comparison with differentiated cell types seems adequate to us. The crux of the question asked by the reviewer is whether the identified ERVs are indeed driving AML or are simply a secondary effect of de-differentiation. It is exactly that

question that we began to answer by performing genetic and epigenetic editing experiments. To clarify our view, we have added the following to the start of the discussion: “Many of these ERVs are also active in CD34+ progenitor cells and are therefore not cancer-specific, but they may nonetheless be used to support a gene expression programme that blocks cellular differentiation, a key hallmark of AML. Genetic and epigenetic perturbation experiments such as the ones presented here, allow us to distinguish between ERVs that support oncogenesis and those whose activation is secondary to cellular de-differentiation.”

4. In the first figure (fig 1B) and associated text, five transposon families appear to be enriched for DHSs in all samples and were therefore eliminated from further analysis. Two of these (TAR1 and MSR1) are satellite repeats and U1 is a snoRNA. These are not transposons and should not have been included in the screening method in the first place. A Blat search shows that most copies of the MER57E3 LTR are in the first intron of ZNF genes, having amplified as part of ZNF gene duplications, so also is not the typical dispersed transposon and should probably also not even be shown. Its DHS enrichment could well be due to proximity to ZNF gene promoters. The last one, LTR13, is also somewhat strange as many copies seem to be located in regions that don't even align with Rhesus (according to a brief search using BLAT).

We apologise if it wasn't clear that we initially analysed the complete Repeatmasker annotation, and not just transposons. Fig. 1B presents all repeat families that came out as significantly enriched in our analysis, after which further selection/curation led us to focus on six ERVs families. We are aware of the nature of U1, TAR1 and MSR1 repeats, as well as the issue with MER57E3 elements, but we did not focus on these families. We made the following adjustments to the text for increased clarity: “We overlapped DNase hypersensitive sites (DHSs) with the complete Repeatmasker annotation and compared the DHS frequency at each repeat family with random controls (Supplementary Table 1). We identified twelve repeat families that were enriched for DHS-associated copies in at least one of the AML cell lines and in 10% or more of the AML samples (Figure 1B). Five of these repeat families (three of which are not TEs) were highly enriched across all samples...”

5. For the data in Figure 1E, which shows the upregulated genes associated with a DHS at a nearby ERV, please provide a supplementary table with the gene names, coordinates of the ERV and the distance of the ERV to the TSS of the gene.

These ERVs/genes were already listed in Supplementary Table 6, which is a compilation of all LTRs with strong evidence of regulatory activity identified in this study. The genes in question are the ones with “DHS-RNA correlation” under the “evidence” column of Supplementary Table 6. We have now added a reference to this table in to the respective section of the main text. We have also changed the table to display the distance to the TSS, as suggested, rather than the distance to any part of the gene (as it was in the previous version). The affected TSS was identified based on RNA-seq data and associated de novo transcriptome assemblies (Blueprint data for DHS-RNA correlation; CRISPRi RNA-seq for the remaining).

6. Related to the above data, the authors discuss three genes at the bottom of page 6 that have some link to AML prognosis. Please elaborate a little more. Is up or down regulation of these genes linked to worse outcome?

We have now expanded this section as follows: “Notably, low *SCIN* expression is associated with an adverse AML prognosis. Two other genes of interest for which expression also correlates with a DHS at nearby ERVs are *TPD52* and *AHSP*, whose overexpression in AML is predictive of poor and favorable outcomes, respectively.”

7. In the section where the authors look for LTRs acting as promoters (page 7), while “promoter effects” seem less important than enhancer effects, At least one gene (RHEX) in the associated Table is probably worth mentioning. LTR-RHEX transcripts are present in a large fraction of AMLs and the LTR2B is one of two annotated promoters in RefSeq. Since this gene is “regulator of hemoglobinization and erythroid cell expansion”, it would be worth commenting on it since it shows up in such a large fraction of AMLs.

We thank the reviewer for highlighting this interesting example. We now mention this example in the results section: “Another example is an LTR2B element that is active in the majority of AML samples and is an annotated promoter of the RHEX gene. RHEX regulates erythroid cell expansion (Verma et al, 2014) and is highly expressed in AML (not shown).”

8. Related to the above, please include a paragraph on how ab initio transcriptome assembly was performed. As well, it is stated in the methods that the assembled transcripts were “visually inspected”. Were these checked against other sources (eg. ESTs or Gencode or FANTOM5) for evidence that the LTRs are indeed the promoters? As an example, for the SUOX gene, the LTR12C seems to provide part of an internal exon in UCSC and is certainly not the promoter.

We have added more details to our methods section, namely that the assembly (as produced by StringTie from HiSat2-aligned RNA-seq data) was guided by GENCODE annotation v26. We have taken the referee's useful suggestion of checking FANTOM5 data, and have added this information to Supplementary Table 2. Many transcripts are not predicted by GENCODE or FANTOM5, which could imply the presence of some artefacts, but

could also constitute previously uncharacterised TSSs. To distinguish between the two would require validation in primary cells, a point we have added to the main text (lines 201-204). Notably, the alternative TSS for *SAGE1* that we show in Figure 2B is not predicted by either GENCODE or FANTOM5, but I hope the referee agrees that this looks very convincing and likely constitutes a bona fide *SAGE1* isoform originating from LTR12C.

In the case of *SUOX*, the picture is less clear, as shown below:

The alternative TSS maps downstream of the exon the referee refers to. It is possible that the signal seen at this putative TSS actually comes from the upstream exon, and that a mappability gap has led StringTie to infer that there is a TSS. However, we cannot rule out the possibility that this is indeed a cryptic internal TSS, which would require validation.

The above considerations reinforce our view that promoter activity is not a major role of A-DARs.

9. Several of the quoted references are not in the reference list. Please fix.

Apologies for not having double checked all the references. We have now fixed this.

Reviewer #3

Figure 1D- were the expression levels averaged across BluePRINT AMLs? This wasn't clear to me.

Yes, these are averages across all samples. We have clarified this in the respective figure legend: "D. Gene expression average across all Blueprint AML samples for genes within 50 kb of A-DARs with or without a DHS in AML and/or in differentiated cells."

Figure 2C- the presence/absence of a particular histone mark at a particular site was called using MACS2 with set thresholds. How were these normalised across samples?

We used processed peak data from Blueprint. Processing details are available at http://dcc.blueprint-epigenome.eu/#/md/chip_seq_grch38, where MACS2 was used with the default q-value cut-off of 0.05 for all samples.

Figure 2F- Indicate more clearly in the figure that the profiles are of H3K27Ac and given the occasionally noisy profiles- mark the peaks that were called by MACS

We have changed the region that is displayed in this figure to one where the signal is clearer. As suggested, we also display the respective peak calls.

Figure 3E- Given the visual association that motifs were enriched at DHS elements, which the authors discount as insignificant- this should be backed up by valid statistics. There is publicly available Hi-C data in K562 cells. It should be possible to directly link the TF bound TEs with gene promoters.

We apologise for the omission of our statistical analysis. We had actually performed this analysis and deposited it on our GitHub repository (https://github.com/MBrancoLab/Deniz_2019_AML), but failed to refer to it on the paper. We have now included the results of this analysis (new Supplementary Data 1) and changed the text accordingly:

"We tested for motif enrichment in elements with DHSs (DHS+) in at least five of the analysed AML samples, when compared to DHS-negative elements (Supplementary Data 1). In four of the ERV families we identified several enriched motifs (none in LTR2C or LTR13A), such as TAL1 (in LTR2B, LTR5_Hs and LTR12C), CEBPB (in LTR2B) and GATA2 (in LTR5B and LTR12C). However, the differences in motif frequency between DHS+ and DHS- elements were modest, making TF motifs poor discriminators of these two groups (Figure 3E)."

I hope it is now clear that we are not denying that there are statistically significant differences, but that those differences are small from a biological standpoint.

Regarding the use of Hi-C data, we gave this issue a lot of attention in the past, but unfortunately the existing Hi-C datasets in K562 do not have the required resolution. To our knowledge, the best dataset is from Rao et al (PMID: 25497547), where a 1 kb resolution is achieved in lymphoblastoid cells. However, the sequencing depth for K562 was lower, such that the identified loops are at 10kb resolution, which is simply not sufficient to resolve promoter-enhancer contacts. There is also ChIA-PET data in ENCODE, but this is of very poor quality.

Figure 4B- Following CRISPR of elements in k562, gene expression values were shown for three ZNF genes. The displayed window in 4A only shows one presumably for practical considerations but it leaves the reader wondering whether RPL7L1 and BIK exist in gene deserts or selective data representation. Use of K562 Hi-C data could potentially show the TE/promoter links. Show 4B as in 5G so the number of clones analysed is evident.

It is indeed for practical reasons that we only provide zoomed in versions of the respective loci. Broader views can be easily accessed through a genome browser. The reason we included the three ZNF genes in figure 4A (top) is that the excision of LTR5B also affects expression of those genes. In contrast, the excision of LTR5B (4A middle) and LTR13A (4A bottom) did not alter expression of other nearby genes, as shown below.

However, we decided not to include these data, as both of these genes have extremely low expression levels, and thus the fact that they are unaffected by the KO has little or no meaning.

Regarding the plots, we haven't shown the data points in Figure 4B, as some experiments have as many as 20 data points. For each experiment, at least 4 replicates (from independent cell harvests) were used from various clones, as detailed in the figure legend. In our opinion, identifying the different clones in individual data points yields a confusing plot that provides little more information, as shown below.

Figure 5- deposition of dCas9 at intronic regions can disrupt transcript elongation independently of disrupting a regulatory element. What proportion of genes with intronic dCas9 (5B) showed reduced expression and could this have impacted on cell growth?

This is an interesting point. We revisited our list of differentially expressed genes and found four with intronic binding of dCas9 (at LTR2/LTR2B elements, i.e., none of them were off-targets): *FLOT2*, *TNS4* and *IL23R* in K562, and *SPIRE1* in OCI-AML3. We cannot exclude the possibility that these genes also impacted on cell growth via interference with transcriptional elongation, and thus we have added a note this effect in the respective results section (lines 364-366). Importantly, we use the CRISPRi experiment to identify promising candidate loci that require independent testing and validation. Our follow-up genetic editing experiments clearly show that LTR2-*APOC1* is a major contributor to the cellular phenotype.

Figure 6- I would be interested to know what the chromatic accessibility/histone profiles at LTR2 adjacent to *APOC1* look like in CD34+?

None of the CD34+ DNase-seq experiments that we analysed displayed a peak at LTR-*APOC1*. We show here one example, where the relevant LTR2 element is highlighted in yellow:

There were a number of references in text that have not transferred to the reference list. This should be corrected.

Apologies for not having double checked all the references. We have now fixed this.

TF binding data is limited to K562 cells. There is publicly available TF data in CD34+ (BloodChIP) Have the authors looked for the presence of TEs at heptad binding sites?

We appreciate the useful pointer from the reviewer. We have now analysed CD34+ data from BloodChIP, and the results are summarised in a new Supplementary Figure 4. Notably, LTR2B does not bind any of these TFs, which is congruent with a lack of DHS enrichment for this ERV family in CD34+ cells. The following text was added (lines 258-261): "To evaluate TF binding in a primary cell type, we analysed data from CD34+ hematopoietic progenitors, from the BloodChIP database. This revealed clear binding enrichment for FLI1, GATA2, LYL1, RUNX1 and TAL1 in at least one of the ERV families (Supplementary Figure 4)."

Regarding heptad binding sites, from the bloodChIP dataset we only identified 153 loci where all seven TFs had peaks (possibly due to lower quality of 1 or 2 ChIP-seq profiles). Only one of these overlaps an A-DAR element, but clearly there are insufficient binding sites to make a thorough assessment.

When focussing on one or other family of LTR from one figure to the next, the authors should justify the selection

It was not clear to us exactly which sections of the manuscript the reviewer was referring to. Our decision to focus on LTR2B for CRISPRi experiments was already justified ("We targeted the LTR2B family, which was the only one with AML-specific DHS enrichment and no enrichment in CD34+ cells (Figure 1B), suggesting a more cancer-specific role than other A-DARs"; lines 331-333). Most other figures present data from all six ERV families of interest, with the exception of Figures 2C and Figure 3E, where only one example is shown for practical reasons. For Figure 2C, results for the remaining families are shown in Supplementary Figure 3B. For Figure 3E, we have now included results for the remaining families in a new Supplementary Figure 5.

Legend 1- line 714- incomplete credit. Line 726- add colours against expression (orange), DHS peak (blue)

Thank you for pointing it out. We have made these changes.

REVIEWERS' COMMENTS:

Reviewer #1 (Remarks to the Author):

Overall the manuscript is now improved. Although the authors did not respond to several of my comments with new experiments, they did provide reasonable explanations since these experiments would be very extensive. I also appreciate the new title as well as the responses to the other reviewers. In my opinion, this is now an excellent manuscript suitable for publication in Nature Communications.

Johan Jakobsson

Reviewer #2 (Remarks to the Author):

The authors have answered all my comments and concerns. It is a very interesting and well done study and I have no further concerns.

Reviewer #3 (Remarks to the Author):

The authors have addressed my concerns and I am satisfied by their responses.